# Heme-binding enables allosteric modulation in an ancient TIM-barrel glycosidase

Gloria Gamiz-Arco[1,8], Luis I. Gutierrez-Rus [1,8], Valeria A. Risso[1], Beatriz Ibarra-Molero [1], Yosuke Hoshino[2], Dušan Petrović[3,7], Jose Justicia [4], Juan Manuel Cuerva [4], Adrian Romero-Rivera[3], Burckhard Seelig [5], Jose A. Gavira [6], Shina C. L. Kamerlin [3✉], Eric A. Gaucher [2✉] & Jose M. Sanchez-Ruiz [1✉]

Glycosidases are phylogenetically widely distributed enzymes that are crucial for the cleavage of glycosidic bonds. Here, we present the exceptional properties of a putative ancestor of bacterial and eukaryotic family-1 glycosidases. The ancestral protein shares the TIM-barrel fold with its modern descendants but displays large regions with greatly enhanced conformational flexibility. Yet, the barrel core remains comparatively rigid and the ancestral glycosidase activity is stable, with an optimum temperature within the experimental range for thermophilic family-1 glycosidases. None of the ~5500 reported crystallographic structures of ~1400 modern glycosidases show a bound porphyrin. Remarkably, the ancestral glycosidase binds heme tightly and stoichiometrically at a well-defined buried site. Heme binding rigidifies this TIM-barrel and allosterically enhances catalysis. Our work demonstrates the capability of ancestral protein reconstructions to reveal valuable but unexpected biomolecular features when sampling distant sequence space. The potential of the ancestral glycosidase as a scaffold for custom catalysis and biosensor engineering is discussed.

[1] Departamento de Quimica Fisica. Facultad de Ciencias, Unidad de Excelencia de Quimica Aplicada a Biomedicina y Medioambiente (UEQ), Universidad de Granada, 18071 Granada, Spain. [2] Department of Biology, Georgia State University, Atlanta, GA 30303, USA. [3] Science for Life Laboratory, Department of Chemistry-BMC, Uppsala University, BMC Box 576, S-751 23 Uppsala, Sweden. [4] Departamento de Quimica Organica. Facultad de Ciencias, Unidad de Excelencia de Quimica Aplicada a Biomedicina y Medioambiente (UEQ), Universidad de Granada, 18071 Granada, Spain. [5] Department of Biochemistry, Molecular Biology, and Biophysics, University of Minnesota, Minneapolis, Minnesota, United States of America, & BioTechnology Institute, University of Minnesota, St. Paul, MN, USA. [6] Laboratorio de Estudios Cristalograficos, Instituto Andaluz de Ciencias de la Tierra, CSIC, Unidad de Excelencia de Quimica Aplicada a Biomedicina y Medioambiente (UEQ), Universidad de Granada, Avenida de las Palmeras 4, Granada 18100 Armilla, Spain. [7] Present address: Hit Discovery, Discovery Sciences, Biopharmaceutical R&D, AstraZeneca 431 50 Gothenburg, Sweden. [8] These authors contributed equally: Gloria Gamiz-Arco, Luis I. Gutierrez-Rus. ✉email: lynn.kamerlin@kemi.uu.se; egaucher@gsu.edu; sanchezr@ugr.es

Pauling and Zuckerkandl proposed in 1963 that the sequences of modern protein homologs could be used to reconstruct the sequences of their ancestors[1]. While this was mostly only a theoretical possibility in the mid-twentieth century, ancestral sequence reconstruction has become a standard procedure in the twenty-first century, due to advances in bioinformatics and phylogenetics, together with the availability of increasingly large sequence databases. Indeed, in the last ~20 years, proteins encoded by reconstructed ancestral sequences ("resurrected" ancestral proteins, in the common jargon of the field) have been extensively used as tools to address important problems in molecular evolution[2,3]. In addition, a new and important implication of sequence reconstruction is currently emerging linked to the realization that resurrected ancestral proteins may display properties that are desirable in scaffolds for enzyme engineering[4–6]. For instance, high stability and substrate/catalytic promiscuity have been described in a number of ancestral resurrection studies[5,7]. These two features are known contributors to protein evolvability[8,9], which points to the potential of resurrected ancestral proteins as scaffolds for the engineering of new functionalities[4,10].

More generally, reconstruction studies that target ancient phylogenetic nodes typically predict extensive sequence differences with respect to their modern proteins. Consequently, proteins encoded by the reconstructed sequences may potentially display altered or unusual properties. Regardless of the possible evolutionary implications, it is of interest, therefore, to investigate which properties of putative ancestral proteins may differ from those of their modern counterparts and to explore whether and how these ancestral properties may lead to new possibilities in biotechnological applications. Here, we apply ancestral sequence reconstruction to a family of well known and extensively characterized enzymes. Furthermore, these enzymes display 3D-structures based on the highly common and widely studied TIM-barrel fold, a fold which is both ubiquitous and highly evolvable[11–13]. Yet, we find upon ancestral resurrection a diversity of unusual and unexpected biomolecular properties that suggest new engineering possibilities that go beyond the typical applications of protein family being characterized.

Glycosidases catalyze the hydrolysis of glycosidic bonds in a wide diversity of molecules[14]. The process typically follows a Koshland mechanism based on two catalytic carboxylic acid residues and, with very few exceptions, does not involve cofactors. Glycosidic bonds are very stable and have an extremely low rate of spontaneous hydrolysis[15]. Glycosidases accelerate their hydrolysis up to ~17 orders of magnitude, being some of the most proficient enzymes functionally characterized[16]. Glycosidases are phylogenetically widely distributed enzymes. It has been estimated, for instance, that about 3% of the human genome encodes glycosidases[17]. They have been extensively studied, partly because of their many biotechnological applications[14]. Detailed information about glycosidases is collected in the public CAZy database (Carbohydrate-Active enZYmes Database; http://www.cazy.org)[18] and the connected CAZypedia resource (http://www.cazypedia.org/)[19]. At the time of our study, glycosidases are classified into 167 families on the basis of sequence similarity. Since perturbations of protein structure during evolution typically occur more slowly than sequences change[20], it is not surprising that the overall protein fold is conserved within each family. Forty eight of the currently described glycosidase families display a fold consistent with the TIM barrel architecture. Often, common ancestry between different TIM-barrel families cannot be unambiguously demonstrated[12]. Therefore, the TIM-barrel may be considered as a "superfold" in the sense of Orengo et al.[21], and simply sharing this fold does not necessarily imply evolutionary relatedness.

Here, we study family 1 glycosidases, which are of the classical TIM-barrel fold. Family 1 glycosidases (GH1) commonly function as β-glucosidases and β-galactosidases, although other activities are also found in the family[22]. GH1 enzymes are present in the three domains of life and have been traced back to LUCA[23]. We focus on a putative ancestor of modern bacterial and eukaryotic enzymes and find a number of unusual properties that clearly differentiate the ancestor from the properties of its modern descendants. The ancestral glycosidase thus displays much-enhanced conformational flexibility in large regions of its structure. This flexibility, however, does not compromise stability as shown by the ancestral optimum activity temperature which is within the typical range for family 1 glycosidases from thermophilic organisms. Unexpectedly, the ancestral glycosidase binds heme tightly at a well-defined site in the structure with concomitant allosteric increase in enzyme activity. Neither metalloporphyrin binding nor allosteric modulation appears to have been reported for any modern glycosidases, despite the fact that these enzymes have been extensively characterized. Overall, this work demonstrates the potential of ancestral reconstruction as a tool to explore sequence space to generate combinations of properties that are unusual or unexpected compared to the repertoire from modern proteins.

## Results

**Ancestral sequence reconstruction**. Ancestral sequence reconstruction (ASR) was performed based on a phylogenetic analysis of family 1 glycosidase (GH1) protein sequences (see the Methods for details). GH1 protein homologs are widely distributed in all three domains of life and representative sequences were collected from each domain, including characterized GH1 sequences obtained from CAZy as well as homologous sequences contained in GenBank. The phylogeny of GH1 homologs consists of four major clades (Fig. 1a and Fig. S1). One clade is composed mainly of archaea and bacteria from the recently proposed Candidate Phyla Radiation (CPR)[24], while the other three clades include bacteria and eukaryotes. The archaeal/CPR clade largely contains uncharacterized proteins and was thus excluded from further analysis. In the bacterial/eukaryotic clades, eukaryotic homologs form a monophyletic clade within bacterial homologs. For our current study, the common ancestors of bacterial and eukaryotic homologs are selected for ASR analysis (N72, N73, and N125) because many homologs have been characterized and there is substrate diversity between the enzymes in the different clades.

**Selection of an ancestral glycosidase for experimental characterization**. We prepared, synthesized, and purified the proteins encoded by the most probabilistic sequences at nodes N72, N73, and N125. While the three proteins were active and stable, we found that those corresponding to N73 and N125 had a tendency to aggregate over time. We therefore selected the resurrected protein from node 72 for exhaustive biochemical and biophysical characterization. For the sake of simplicity, we will subsequently refer to this protein as the ancestral glycosidase in the current study.

It is important to note that the sequence of the ancestral glycosidase differs considerably from the sequences of modern proteins. The set of modern sequences used as a basis for ancestral reconstruction span a range of sequence identity (26–59%) with the ancestral glycosidase (Table S1). Also, using the ancestral sequence as the query of a BLAST (Basic Local Alignment Search Tool) search in several databases (non-redundant protein sequences, UniProtKB/Swiss-Prot, Protein Data Bank, Metagenomic proteins) yields a closest hit with only a 62% sequence identity to the ancestral glycosidase. These sequence differences translate into unexpected biomolecular properties.

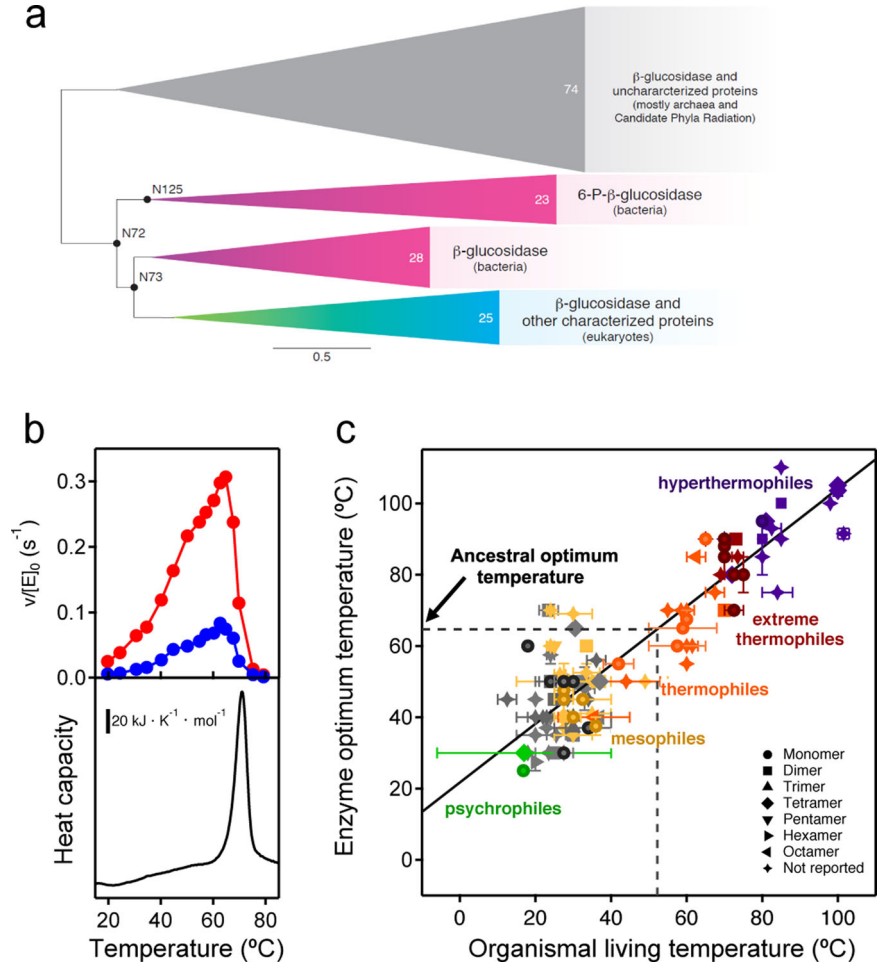

**Fig. 1 Ancestral sequence reconstruction of family 1 glycosidases (GH1) and assessment of ancestral stability. a** Bayesian phylogenetic tree of GH1 protein sequences using 150 representative sequences. Triangles correspond to four major well-supported clades (see supplemental Fig. S1 for nodal support) with common functions indicated. Numbers inside the triangles correspond to the number of sequences in each clade. Scale bar represents 0.5 amino acid replacements per site per unit evolutionary time. Reconstructed ancestral sequences were inferred at the labeled nodes and the protein at node 72 was exhaustively characterized. **b** Determination of the optimum temperature for the ancestral glycosidase (upper panel) using two different substrates 4-nitrophenyl-β-D-glucopyranoside (red) and 4-nitrophenyl-β-D-galactopyranoside (blue). $v/[E]_0$ stands for the rate over the total enzyme concentration. The lower panel shows a differential scanning calorimetry profile for the ancestral glycosidase. Clearly, the activity drop observed at high temperature (upper panel) corresponds to the denaturation of the protein, as seen in the lower panel. **c** Plot of enzyme optimum temperature versus living temperature of the host organism for modern family 1 glycosidases. Data (Supplementary Dataset 1) are derived from literature searches, as described in Methods. Horizontal and vertical bars are not error bars, but represent ranges of organismal living temperatures and enzyme optimum temperatures when provided in the literature. Color code denotes the organisms that published literature describes as hyperthermophiles, extreme thermophiles, thermophiles, mesophiles, psychrophiles; gray color is used for organisms that have not been thus classified (plants that live at moderate temperatures in most cases). The line is a linear-squares fit ($T_{OPT} = 21.68 + 0.824 T_{LIVING}$). Correlation coefficient is 0.89 and $p \sim 8.8 \times 10^{-45}$ (probability that the correlation results from chance). An environmental temperature of about 52 °C can be estimated from the optimum temperature of the ancestral glycosidase.

**Stability**. As it is customary in the glycosidase field, we assessed the stability of the ancestral glycosidase using profiles of activity versus temperature determined by incubation assays[25]. These profiles typically reveal a well-defined optimum activity temperature (Fig. 1b) as a result of the concurrence of two effects. At low temperatures, the expected Arrhenius-like increase of activity with temperature is observed. At high temperatures, protein denaturation occurs and causes a sharp decrease in activity. For the ancestral glycosidase, this interpretation is supported by differential scanning calorimetry data (lower panel in Fig. 1b) which show a denaturation transition that spans the temperature range in which the activity drops sharply.

The profiles of activity versus temperature (Fig. 1b) show a sharp maximum at 65 °C for the optimum activity temperature of the ancestral glycosidase. In order to ascertain the implications of this value for the evaluation of the ancestral stability, we have searched the literature on family 1 glycosidases for reported optimum temperature values (see Methods for details and Supplementary Dataset 1 for the results of the search). The values for ~130 modern enzymes show a good correlation with the environmental temperature of their respective host organisms (Fig. 1c). Therefore, the enzyme optimum temperature is an appropriate reflection of stability in an environmental context for this protein family. The optimum temperature value for ancestral glycosidase is within the experimental range of optimum activity temperatures for family 1 glycosidases from thermophilic organisms and it is consistent with an ancestral environmental temperature of about 52 °C (Fig. 1c).

**Conformational flexibility**. Remarkably, despite its "thermophilic" stability, large regions in the structure of the ancestral

glycosidase are flexible and/or unstructured, as demonstrated by both experiment and computation (Fig. 2).

Proteolysis is known to provide a suitable probe of conformational diversity and the protein energy landscape[26], since most cleavable sites are not exposed in folded compact protein states. The ancestral glycosidase is highly susceptible to proteolysis and degradation is already apparent after only a few minutes incubation at a low concentration of thermolysin (0.01 mg/mL,

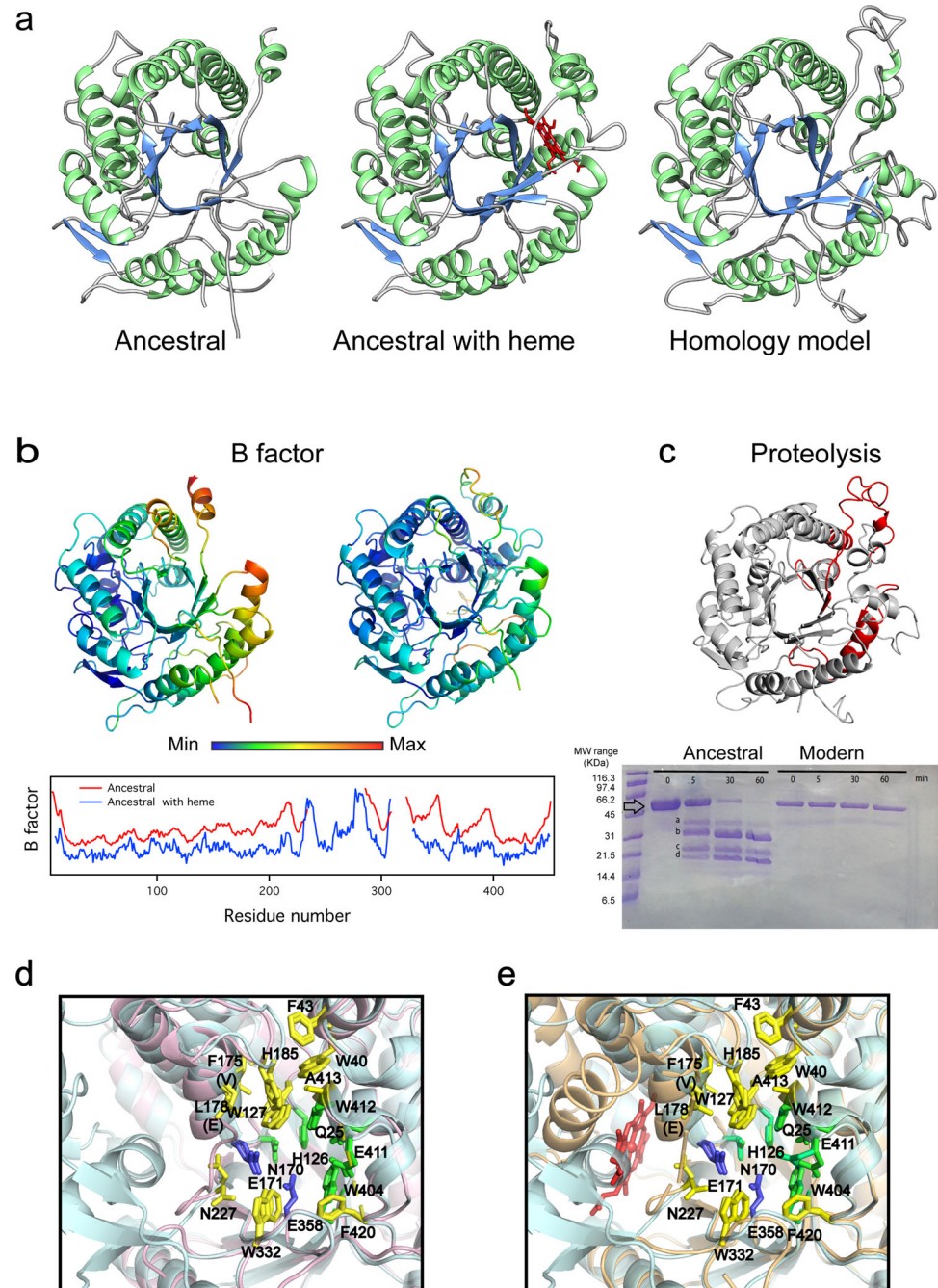

**Fig. 2 3D Structure of the ancestral glycosidase as determined by X-ray crystallography. a** Comparison between the ancestral structure determined in the absence (left) and presence (middle) of bound heme (red) and a homology model constructed as described in Supporting Information. The visual comparison reveals the missing sections in the electronic density of the ancestral protein, mostly in the protein without heme bound. **b** 3D structure of the ancestral protein without and with heme bound color-labeled according to normalized B-factor value and profiles of normalized B-factor versus residue number for the ancestral protein without (red) and with (blue) bound heme. Values are not shown for the sections that are missing in the experimental structures. **c** Proteolysis experiments with the ancestral glycosidase and the modern glycosidase from *Halothermothrix orenii*. The major fragments are labeled *a*, *b*, *c* and *d*. Molecular weights (MW) are shown for the markers used. Five independent experiments were performed with similar results. Mass spectrometry of the fragments predicts cleavage points within the red labeled sections in the shown structure. **d** Superposition of the structure of the ancestral glycosidase with that of the modern glycosidase from *Halothermothrix orenii* showing the critical active-site residues. **e** Superposition of the structures of the ancestral glycosidase without and with heme bound showing the critical active-site residues. In both (**d**) and (**e**), the highlighted active-site residues include the catalytic carboxylic acid residues (blue) and the residues involved in binding of the glycone (yellow) and aglycone (green) parts of the substrate.

Fig. 2c, and Fig. S2). Conversely, the modern glycosidase from the thermophilic *Halothermothrix orenii* remains essentially unaffected after several hours with the same concentration of the protease (Fig. 2c) or even with a ten times larger protease concentration. These two glycosidases, modern thermophilic and putative ancestral, are monomeric, as determined from gel filtration chromatography and analytical ultracentrifugation (Figs. S3 and S4) and display similar values for the optimum activity temperature (70 °C and 65 °C, respectively: Fig. 1b and S5). Therefore, their disparate susceptibilities to proteolysis can hardly be linked to differences in overall stability, but rather to enhanced conformational flexibility in the ancestral enzyme that exposes cleavable sites.

Furthermore, there is a large region missing in the electronic density map of the ancestral protein from X-ray crystallography (Fig. 2a), while the rest of the model agrees with a homology model based on modern glycosidase structures (see Supplementary Methods for details). At the achieved resolution of 2.5 Å, it should be possible to trace the course of a polypeptide chain in space, provided that such course is well defined. Therefore, the missing regions very likely correspond to regions of high flexibility. In addition, flexibility is also suggested by the B-factor values in regions that are present in the ancestral structure (Fig. 2b).

Lastly, molecular dynamics (MD) simulations (Fig. 3) also indicate enhanced flexibility in specific regions as shown by cumulative 15 μs simulations of the substrate-free forms of the ancestral glycosidase (both with and without heme: see below) as well as the modern glycosidase from *Halothermotrix orenii* (PDB ID: 4PTV)[27] [https://www.rcsb.org/structure/4PTV]. Both ancestral and modern proteins have the same sequence length,

and similar protein folds with a root mean square deviation (RMSD) difference of only 0.7 Å between the structures. However, our molecular dynamics simulations indicate a clear difference in flexibility in the region spanning residues 227–334, which is highly disordered in the ancestral glycosidase but ordered and rigid in the modern glycosidase, with root mean square fluctuation (RMSF) values of <2 Å (Fig. 3). We also analyzed the interactions formed between residues 227–334 and the rest of the protein by counting the total intramolecular hydrogen bonds formed along the MD simulations. We observe that on average, the modern glycosidase forms $115 \pm 11$ hydrogen bonding interactions during our simulations, whereas the ancestral glycosidase forms either $101 \pm 12/101 \pm 13$ hydrogen bonding interactions (in the presence of heme and absence of heme, respectively). This suggests that the higher number of intramolecular hydrogen bonds formed between residues 227–334 and the protein can contribute to the reduced conformational flexibility observed in the case of the modern glycosidase, compared to the ancestral glycosidase.

It is important to note that there is a clear structural congruence between the results of the experimental and computational studies described above. That is, the missing regions in the X-ray structure (Fig. 2a) match the high-flexibility regions in the molecular dynamics simulations (Fig. 3) and include the proteolysis cleavage sites determined by mass spectrometry (Fig. 2c). Overall, regions encompassing two alpha helices and several loops appear to be highly flexible or even unstructured in the ancestral glycosidase. The barrel core, however, remains structured and shows comparatively low conformational flexibility, which may explain the high thermal stability of the protein (see Discussion).

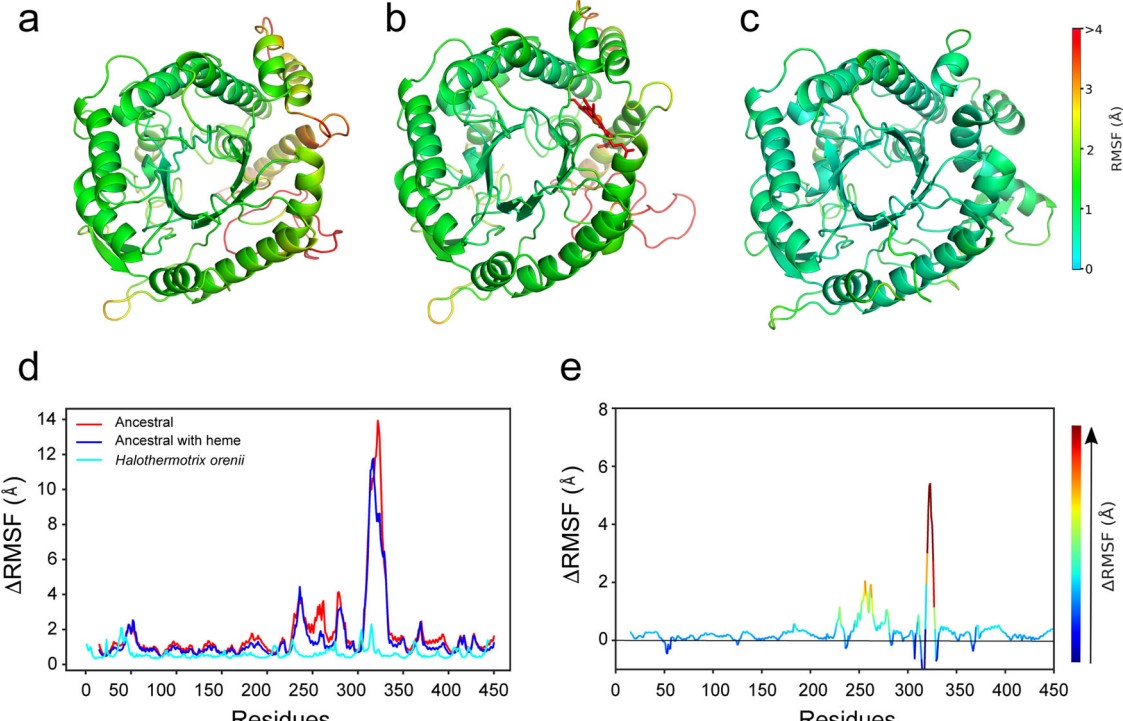

**Fig. 3 Molecular dynamics simulations.** Representative snapshots from molecular dynamics simulations of ancestral and modern glycosidases, showing the ancestral glycosidase both (**a**) without and (**b**) in complex with heme, as well as (**c**) the corresponding modern protein from *Halothermothrix orenii*. Structures were extracted from our simulations based on the average structure obtained in the most populated cluster using the hierarchical agglomerative algorithm implemented in CPPtraj[64]. All protein structures are colored by calculated root mean square fluctuations (RMSF) over the course of simulations of each system (see the color bar). Shown are also (**d**) absolute and (**e**) relative RMSF (Å) for each system, in the latter case showing the RMSF of the ancestral glycosidase without heme relative to the heme bound structure. Note the difference in the color bars between panels (**a**–**c**), which describes absolute RMSF per system, and panel (**e**), which describes relative RMSF. The numerical scale of the color bar on panel (**e**) corresponds to the *y*-axis of this panel.

**Catalysis**. We determined the Michaelis–Menten parameters for the ancestral enzyme with the substrates typically used to test the standard β-glucosidase and β-galactosidase activities of family 1 glycosidases (4-nitrophenyl-β-D-glucopyranoside and 4-nitrophenyl-β-D-galactopyranoside) (Fig. 4 and Tables S2 and S3). We also compared the results with the catalytic parameters for four modern family 1 glycosidases, specifically those from *Halothermothrix orenii*, *Marinomonas sp.* (strain MWYL1), *Saccharophagus degradans* (strain 2-40 T), and *Thermotoga maritima* (Figs. S6–S9 and Tables S2 and S3). Modern glycosidases are highly proficient enzymes accelerating the rate of glycoside bond hydrolysis up to about 17 orders of magnitude[16]. The ancestral

enzyme appears to be less efficient and shows a turnover number about two orders of magnitude below the values for the modern glycosidases studied here (Fig. 4). It is important to note, nevertheless, that the turnover number for the ancestral enzyme is ~13 orders of magnitude higher than the first-order rate constant for the uncatalyzed hydrolysis of β-glucopyranosides, as determined by Wolfenden through Arrhenius extrapolation from high-temperature rates[15]. The catalytic carboxylic acid residues as well as the residues known to be responsible for the interaction with the glycone moiety of the substrate[28] are present in the ancestral enzyme and appear in the static X-ray structure in a configuration similar to that observed in the modern proteins

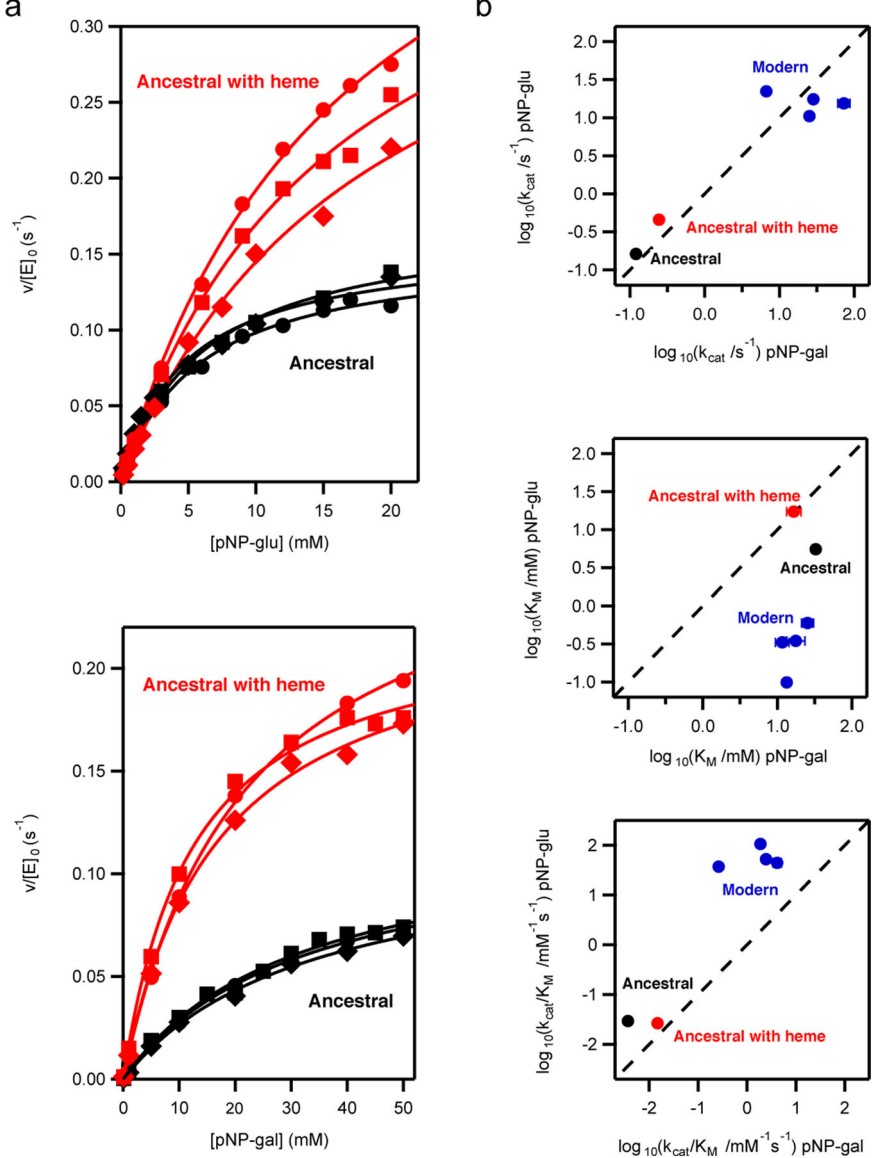

**Fig. 4 Ancestral versus modern catalysis by family 1 glycosidases. a** Michaelis plots of rate versus substrate concentration at pH 7 and 25 °C for hydrolysis of 4-nitrophenyl-β-D-glucopyranoside (upper panel) and 4-nitrophenyl-β-D-galactopyranoside (lower panel) catalyzed by the ancestral glycosidase with and without heme bound. v/[E]$_0$ stands for the rate over the total enzyme concentration. The lines are the best fits of the Michaelis–Menten equation. The different symbols (diamond, square, circle) refer to the triplicate experiments (involving two different protein preparations) performed for each protein/substrate combination. Michaelis plots for the four modern proteins studied in this work can be found in Figs. S6–S9. The values for the catalytic parameters derived from these fits are collected in Tables S2 and S3. **b** Logarithm of the Michaelis-Menten catalytic parameters for a glucopyranoside substrate versus a galactopyranoside substrate. pNP-glu and pNP-gal stand, respectively, for 4-nitrophenyl-β-D-glucopyranoside and 4-nitrophenyl-β-D-galactopyranoside. $k_{cat}$, $K_M$, and $k_{cat}/K_M$ stand for the turnover number, the Michaelis constant and the catalytic efficiency. The values shown are averages of the values derived from the triplicates and the associated errors are the corresponding standard deviations. Note that, in most cases, the associated errors are smaller than the size of the data points.

(Fig. 2d). There are a few differences in the identity of the residues responsible for the binding of the aglycone moiety of the substrate[28], but these differences occur in positions that are variable in modern family 1 glycosidases (Fig. S10 and Table S4). Overall, the comparatively low activity of the ancestral protein is likely linked to its conformational flexibility. That is, the protein in solution is sampling a diversity of conformations of which only a few are active towards the common substrates. From an evolutionary point of view, the comparatively low ancestral activity may reflect an early stage in the evolution of family 1 glycosidases before selection favored greater turnover (see "Discussion").

Also, it is interesting to note that, although both β-glucosidase and β-galactosidase activities are typically described for family 1 glycosidases, these enzymes are commonly specialized as β-glucosidases[22]. This specialization does not occur, however, at the level of the turnover number, which is typically similar for both kinds of substrates. Instead, specialization occurs at the level of the substrate affinity, as reflected in lower values of the Michaelis constant ($K_M$) for β-glucopyranoside substrates as compared to β-galactopyranoside substrates[22]. This pattern is indeed observed in the modern enzymes we have studied (Fig. 4), which are described in the literature as β-glucosidases. On the other hand, this kind of specialization is not observed in the ancestral glycosidase, which shows similar $K_M$'s for the β-glucopyranoside and the β-galactopyranoside substrates. This lack of specialization may again reflect an early stage in the evolution of family 1 glycosidases, an interpretation which would seem generally consistent with the fact that resurrected ancestral proteins often display promiscuity[5,7,9,29]. On the other hand, it can be argued that the ancestral glycosidase was specialized for a different kind of substrate. To explore this possibility, we determined catalytic rates for a wide range of glycosidase substrates. These studies are briefly described below:

(1) Using the same methodology employed with 4-nitrophenyl-β-D-glucopyranoside and 4-nitrophenyl-β-D-galactopyranoside (Fig. 1), we determined profiles of catalytic rate versus temperature for the ancestral glycosidase and the modern glycosidases from *Halothermothrix orenii* and *Saccharophagus degradans* using as substrates 4-nitrophenyl-β-D-fucopyranoside, 4-nitrophenyl-β-D-lactopyranoside, 4-nitrophenyl-β-D-xylopyranoside and 4-nitrophenyl-β-D-mannopyranoside. In all cases (Fig. S11), we found the levels of catalysis of the ancestral protein to be reduced in comparison with the modern proteins. We also found that the levels of catalysis for the β-glucopyranoside and β-fucopyranoside substrates were similar, but this pattern is also observed with the modern proteins. (2) We carried out single activity determinations at 25 °C for the ancestral glycosidase with a wider range of substrates, including derivatives of disaccharides (maltose, cellobiose) and several substrates with an α anomeric carbon (Table S5). However, we did not find any substrate with a catalysis level substantially higher than that of those previously determined for 4-nitrophenyl-β-D-glucopyranoside and 4-nitrophenyl-β-D-galactopyranoside and, in many cases (in particular with the α substrates), no substantial activity was detected. (3) Since some of the proteins that descended from the N72 node are 6-phosphate-β-glucosidases (Fig. 1A and S1), we tested the activity of our ancestral glycosidase against 4-nitrophenyl-β-D-glucopyranoside-6-phosphate (Fig. S12). We found the catalytic efficiency to be ~40 fold smaller than that determined with the corresponding non-phosphorylated substrate. (4) Glycosidases are typically described[14] as being very promiscuous for the aglycone moiety of the substrate (the part of the substrate that is replaced with *p*-nitrophenyl in the substrates commonly used to assay glycosidase activity) while they are more specialized for the glycone moiety of the substrate. However, the flexibility in certain regions of the ancestral structure could perhaps favor the hydrolysis of substrates with larger aglycone moieties. To explore this hypothesis, we tested four synthetic

substrates with aglycone moieties larger than the usual *p*-nitrophenyl group (Fig. S13). We revealed that ancestral levels of catalysis are substantially reduced with respect to those obtained for the modern glycosidase from *Halothermothrix orenii*, used here as comparison.

**Heme binding and allosteric modulation.** Overall, it appears reasonable that our resurrected ancestral enzyme reflects an early stage in the evolution of family 1 glycosidases, perhaps following a fragment fusion event (see Discussion), at which catalysis was not yet optimized and substrate specialization had not yet evolved. The presence of a large unstructured and/or flexible regions in the ancestral structure could perhaps reflect the absence of a small molecule that binds within that region. While these proposals are speculative, the experimental results described in detail below, show that the ancestral glycosidase does bind heme tightly and stoichiometrically at a site in the flexible regions. This was a completely unexpected observation given the large number of modern glycosidases that have been characterized in the absence of any porphyrin rings.

We curiously noticed that most preparations of the ancestral glycosidase showed a light-reddish color after elution from an affinity column (Fig. 5). UV–Vis spectra revealed the pattern of bands expected for a heme group[30], including the Soret band at about 400 nm and, in some cases, even the weaker α and β bands (i.e., the Q bands) in the 500–600 nm region (Fig. 6). From the intensity of the Soret band, a very low heme:protein ratio of about 0.02 was estimated for standard enzyme preparations, indicating that all the experiments described above were performed with essentially heme-free protein. However, the amount of bound heme in protein preparations was substantially enhanced by including hemin in the culture medium (heme with iron in the +3 oxidation state) or 5-aminolevulinic acid, the metabolic precursor of heme (Fig. 5). Heme:protein ratios of about 0.10 and 0.18, respectively, were then obtained (Fig. 6a). These results suggest that the ancestral glycosidase does have the capability to bind heme, but also that, as is commonly the case with modern heme-binding proteins[31], the limited amount of heme available in the expression host, combined with the high protein over-expression levels used, leads to low heme:protein ratios. The capability of the ancestral enzyme to bind heme was first shown by the in vitro experiments described next, and then confirmed by mass spectrometry and X-ray crystallography as subsequently described.

Heme has a tendency to associate in aqueous solution at neutral pH, a process that is reflected in a time-dependent decrease in the intensity of the Soret band, which becomes "flatter" upon the formation of dimers and higher associations[32]. However, the process is reversed upon addition of the essentially heme-free ancestral glycosidase (Fig. 6b), indicating that the protein binds heme and shifts the association equilibria towards the monomeric state. Remarkably, heme binding is also reflected in a several-fold increase in enzymatic activity which occurs on the seconds time scale when the heme and enzyme concentration are at a ~micromolar concentration (Fig. 6c). Determination of activity after suitable incubation times for different heme:protein ratios in solution yielded a plot with an abrupt change of slope at a stoichiometric ratio of about 1:1 (Fig. 6d). These experiments were carried out with ~micromolar heme and protein concentrations, indicating therefore a tight, sub-micromolar binding. This interaction was confirmed by microscale thermophoresis experiments that yielded an estimate of 547±110 nM for the heme dissociation constant (see "Methods" and Fig. S14 for details). Indeed, in agreement with this tight binding, increases in activity upon heme addition to a protein solution were observed (Fig. 6c) even with concentrations of ~50 nanomolar.

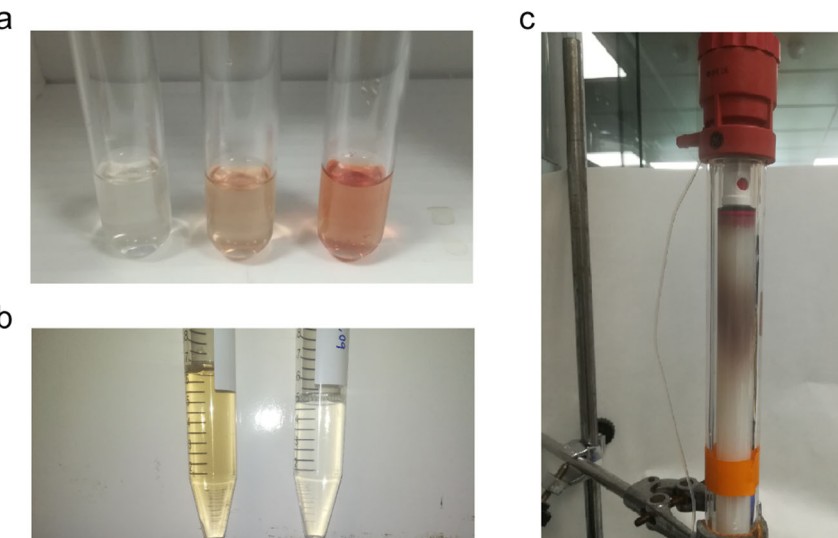

**Fig. 5 Heme binding to the ancestral glycosidase is visually apparent. a** The ancestral protein was prepared by Ni-NTA affinity chromatography. The pictures show the samples eluted from the columns for three different preparations that differed by the addition of 20 μM hemin (middle) and 0.4 mM 5-aminolevulinic acid (right) to the culture medium. Neither hemin nor 5-aminolevulinic acid had been added to the culture medium in the preparation on the left. Protein concentrations in these samples were ∼10 mg/mL. **b** An ancestral glycosidase sample with a low amount of bound heme (right) was incubated with an excess of heme. A PD10 column and FPLC (fast protein liquid chromatrography) were then used to remove the unbound heme. The resulting protein preparation is shown on the left. Protein concentration is ∼0.5 mg/mL. **c** The position of the ancestral protein with bound heme in a FPLC column is revealed by a reddish-brown band.

The 1:1 stoichiometry of heme/protein was confirmed by experiments in which the protein was incubated with an excess of heme and free heme was removed through exclusion chromatography (2 passages through PD10 columns). The protein was then quantified by the bicinchoninic acid method[33] with the Pierce™ BCA Protein Assay Kit while the amount of heme was determined using the pyridine hemochrome spectrum[34] after transfer to concentrated sodium hydroxide (see Methods for details). This resulted in a heme/protein stoichiometry of 1.03±0.03 from five independent assays.

The experiments described above allowed us to set up a procedure for the preparation of the ancestral protein saturated with heme and to use this preparation for activity determinations and crystallization experiments. The procedure (see Methods for details) involved in vitro reconstitution using hemin but did not include any chemical system capable of performing a reduction. It is therefore safe to assume that our heme-bound ancestral glycosidase contains iron in the +3 oxidation state. Activity determinations with the heme-saturated ancestral enzyme corroborated that heme binding increases activity by ∼3 fold (see Michaelis plots in Fig. 4). Both mass spectrometry (Fig. S15) and X-ray crystallography confirmed the presence of one heme per protein molecule (Fig. 7, S16 and S17), which is located at the same site, with the same orientation and involved largely in the same molecular interactions in the three protein molecules (A, B, C) observed in the crystallographic unit cell. Besides interactions with several hydrophobic residues, the bound heme interacts (Fig. 7a) with Tyr264 of α-helix 8 (as the axial ligand), Tyr350 of α-helix 13, Arg345 of β-strand B and, directly via a water molecule, with Lys 261 of β-strand B, although this latter interaction is only observed in chain A. The bound heme shows B-factor values similar to those of the surrounding residues (Fig. 7b), it is well-packed and 95% buried (Fig. 7c). Indeed, the accessible surface area of the bound heme is only 43 Å² compared to the ∼800 Å² accessible surface area for a free heme[35]. The interactions of the bound heme in the ancestral glycosidase are overall similar to those described for modern b-type heme

proteins[35–37]. As observed in modern heme-binding proteins, the ancestral heme-binding pocket is enriched in hydrophobic and aromatic residues and propionate anchoring is achieved through interactions with arginine, tyrosine and lysine residues. Certainly, tyrosine, the axial ligand in the ancestral glycosidase, is not the most common axial ligand in modern heme proteins, but it is found in several cases, including catalases (see, Protein Data Bank (PDB) ID 1QWL for the 3D structure of the catalase from *Helicobacter pylori*). Interestingly, the amino acid residues that interact with the heme in the ancestral glycosidase are somewhat conserved, and are indeed the consensus residue from the set of modern glycosidases used as the starting point for ancestral reconstruction (Table S6). The fraction of each consensus residues in the modern protein is, however, less than unity and the sequences of modern glycosidases in the set differ from the ancestral sequence at many of the positions involved in heme interactions in the ancestral protein (Table S7).

Heme binding clearly rigidifies the ancestral protein, as shown by fewer missing regions in the electronic density map, in contrast to the structure of the heme-free protein (see Fig. 2a–c). This is also confirmed by molecular dynamics (MD) simulations of the ancestral glycosidase both with and without heme bound (Fig. 3 and S18). Figure S18 shows the backbone RMSD (root mean square deviation) over ten individual 500 ns MD simulations per system, and, from this data, it can be seen that while the RMSD is fairly stable in the case of the modern protein, the ancestral glycosidases (both with and without heme bound) are initially quite far from their equilibrium structures, due to the high flexibility of the missing regions of the protein which require substantial equilibration. In addition, we note that while the overall average RMSD for the ancestral protein with heme bound is slightly lower than for the ancestral protein without heme (Fig. S18), the standard deviation is higher. This is due to the greater flexibility of the reconstructed missing loop (see the "Methods" section), which allows it to sample a larger span of conformations depending on whether the loop is interacting with the bound heme or not (we observe both scenarios in our simulations of the heme-bound ancestral glycosidase).

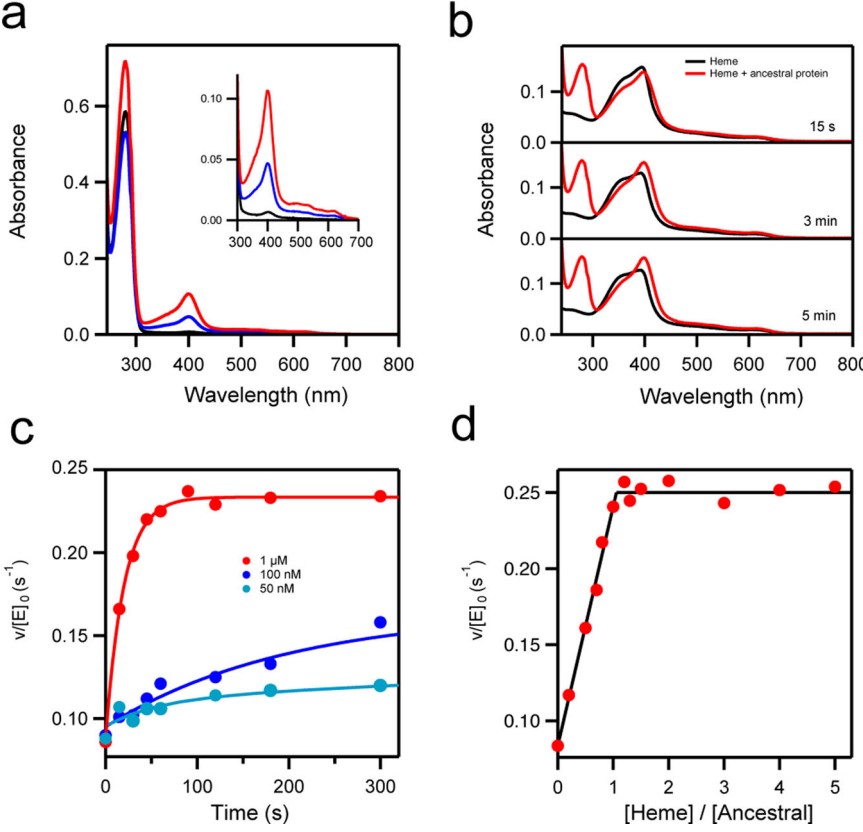

**Fig. 6 Heme binding to the ancestral glycosidase. a** UV–VIS spectra for preparations of the ancestral glycosidase showing the protein absorption band at about 280 nm and the absorption bands due to the heme (the Soret band at about 400 nm and the Q bands at higher wavelengths). Black color is used for the protein obtained using the original purification procedure without the addition of hemin or hemin precursor. Blue and red are used to refer, respectively, to preparations in which hemin and 5-aminolevulinic acid (the metabolic precursor of heme) were added to the culture medium. **b** Binding of heme to the ancestral glycosidase in vitro as followed by changes in VIS spectrum. Spectra of a heme solution 1 μM in the absence (black) or presence (red) of a similar concentration of ancestral protein. The "flat" Soret band of free heme is linked to its self-association in solution, while the bound heme is monomeric and produces a sharper Soret band. **c** Kinetics of binding of heme to the ancestral glycosidase as followed by the increase in enzyme activity (rate of hydrolysis of *4*-nitrophenyl-β-glucopyranoside; see Methods for details). In the three experiments shown a heme to protein molar ratio of 1.2 was used. The protein concentration in each experiment is shown. Note that activity increase is detected even with concentrations of 50 nM, indicating that binding is strong. The lines are meant to guide the eye and thus have no quantitative purpose. (**d**) Plot of enzyme activity versus [heme]/[protein] ratio in solution for a protein concentration of 1 μM. Activity was determined after a 5 min incubation and the plot supports a 1:1 binding stoichiometry. $v/[E]_0$ stands for the rate over the total enzyme concentration.

In contrast, in the absence of the heme, the loop is always in a flexible open conformation leading to a higher overall RMSD but a lower standard deviation as a narrower range of conformations are sampled in our simulations. As neither the loop nor the heme has access to the active site (Fig. S17), these differences are unlikely to have a direct effect on catalysis.

The higher flexibility of the ancestral protein without heme bound can also be seen from comparing RMSF (root mean square fluctuation) values across the protein. That is, the MD simulations performed without the heme bound show that most of the protein has higher flexibility (Fig. 3e, with ΔRMSF values greater than 0 in most of the sequence), particularly in the regions where the B-factors also indicate high flexibility (Fig. 2b). This is noteworthy, as the only difference in starting structure between the two sets of simulations is the presence or absence of the heme; the starting structures are otherwise identical. The MD simulations show that removing the heme from the heme-bound structure has a clear effect on the flexibility of the whole enzyme, increasing it relative to the heme bound structure (Fig. 3e), again also indicated by the B-factors (Fig. 2b). There are two regions where this difference is particularly pronounced. The first spans residues 25–265, which is located

where the heme Fe(III) atom forms an interaction with the Tyr264 side chain as an axial ligand. Removing the heme removes this interaction, thus inducing greater flexibility in this region. The second region with increased flexibility spans 319–327, where again we observe that removing the heme increases the flexibility of this region.

Lastly, we note that the heme is located near the enzyme active site (at about 8 Å from the catalytic glutamate at position 171) but does not have direct access to this site as revealed in the structure (Fig. S17). Therefore, the increase in activity observed upon heme binding is an allosteric effect likely linked to dynamics (see "Discussion") since heme binding does not substantially alter the position/conformation of the catalytic carboxylic acids nor the residues involved in substrate binding according to the static X-ray structures (Fig. 2e). In fact, examining backbone RMSF values of key catalytic residues (Fig. S19) indicates that the flexibility of several of these residues is reduced upon moving from the ancestral glycosides without heme, to adding the heme, to the modern glycosidase, in a clear decreasing trend. We note that the observed effects are subtle and sub-Å; however, there are several experimental studies that suggest that sub-Å changes in dynamics can be catalytically important[38–40].

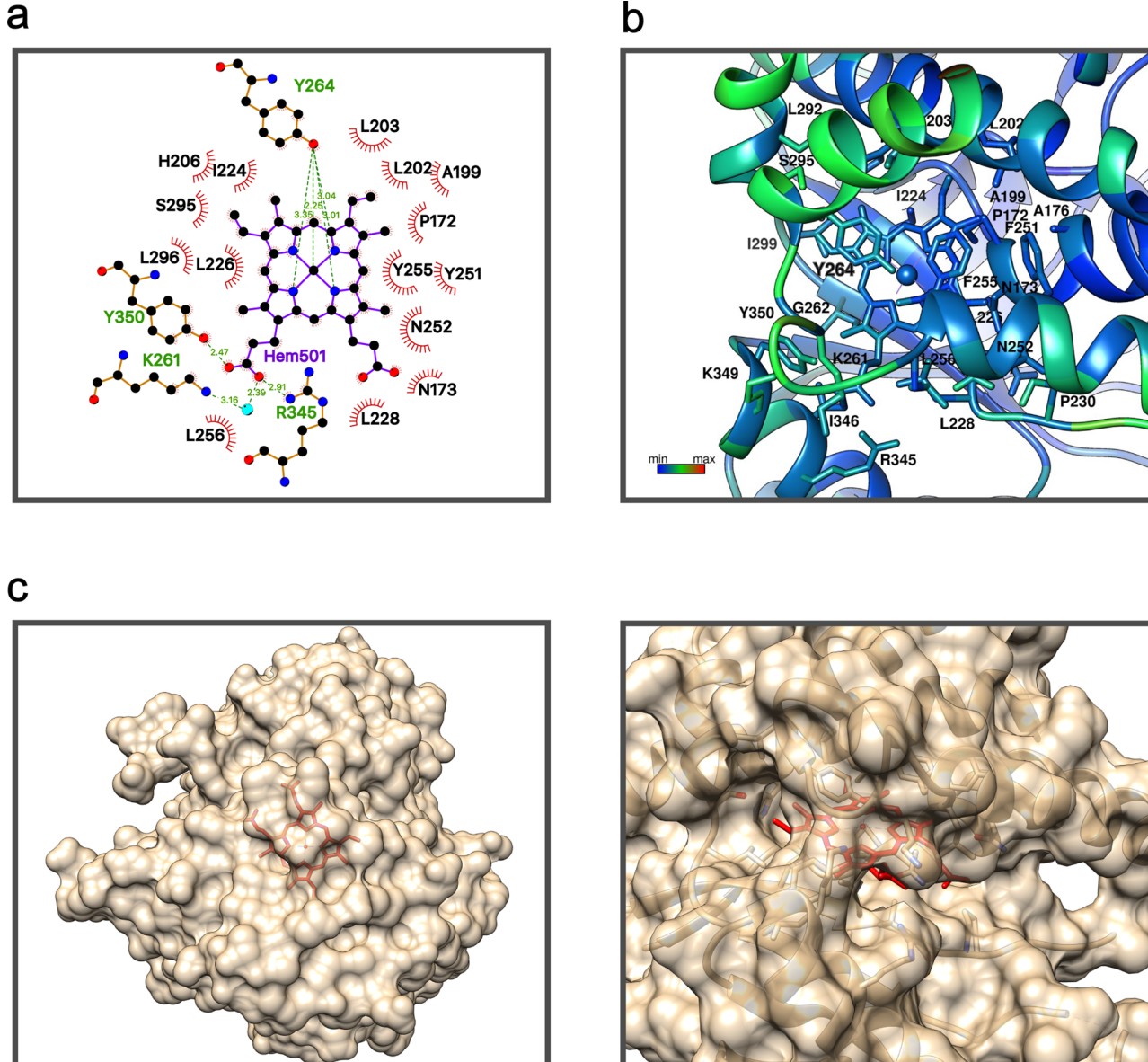

**Fig. 7 Local molecular environment of the bound heme in the ancestral protein. a** Schematic representation of the heme molecule and the neighbor residues in the 3D structure. **b** Heme group and residues directly interacting with the heme colored by B value. **c** Van der Waals surface of the ancestral protein shown in translucent brown, so that it becomes visually apparent that the heme (shown in red) is mostly buried.

## Discussion

The TIM-barrel is the most common enzyme fold, accounting for ~10% of known enzyme structures and providing a scaffold for an enormous diversity of biomolecular functions[11–13]. It is composed of eight parallel (β/α) units linked by hydrogen bonds forming a cylindrical core ("the barrel") with secondary structure elements connected by loops. The high capability of the fold to accommodate a wide diversity of different natural functions is likely linked to its modular architecture, with the barrel (and the αβ loops) providing stability and allowing a substantial degree of flexibility, variability, and, therefore, evolvability for the βα loops. That is, the barrel provides a stable platform that can accommodate loops of different sequences and conformations at the so-called catalytic face.

Remarkably, the differences in conformational flexibility between different parts of the molecule appear to be even more pronounced in our ancestral TIM-barrel glycosidase. Stability is still guaranteed by a rigid barrel core, but flexibility is

greatly enhanced and extends to large parts of the structure, as shown by a combination of computational and experimental results. Conformational flexibility implies that the protein in solution is sampling a diversity of conformations. On the one hand, this may prevent the enzyme from reaching the highest levels of catalysis for a given natural reaction since the protein ensemble may not be shifted towards the most active conformations. Indeed, while modern glycosidases approach catalysis levels up to 17 orders of magnitude above the rate of spontaneous glycoside bond hydrolysis[16], the ancestral glycosidase displays turnover numbers about two orders of magnitude below the modern glycosidases studied here (Fig. 4 and Tables S2 and S3). On the other hand, flexibility is key to the emergence of new functions and contributes to evolvability, since minor conformations that catalyze alternative reactions may be enriched by subsequent evolution[41–44]. Therefore, the ancestral TIM-barrel described here holds promise as a scaffold for the generation of de novo catalysts, an important and largely unsolved problem in

enzyme engineering. We have recently shown[45] that completely new enzyme functions can be generated through a single mutation that generates both a cavity and a catalytic residue, provided that conformational flexibility around the mutation site allows for substrate and transition-state binding[10,43,44]. The combination of a rigid core that provides stability with high flexibility in specific regions makes the ancestral protein studied here an excellent scaffold to develop this minimalist approach to de novo catalysis (work in progress).

Catalytic features of the ancestral glycosidase, such as diminished activity levels and lack of specialization for glucopyranoside substrates, would seem consistent with an early stage in the evolution of family 1 glycosidases. It has been proposed that TIM-barrel proteins originated through fusions of smaller fragments[46]. The high conformational flexibility in some regions of the ancestral glycosidase structure would then also seem consistent with an early evolutionary stage, since fragment fusion is not expected to immediately lead to efficient packing and conformational rigidity in all parts of the generated structure. On the other hand, the capability of the reconstructed ancestral glycosidase to bind heme tightly and stoichiometrically at a well-defined site is rather surprising. None of the ~5500 X-ray structures for the ~1400 glycosidases currently reported in CAZy shows a porphyrin ring. It is certainly possible that heme binding to the ancestral glycosidase is simply an accidental byproduct of the high conformational flexibility at certain regions of the structure, although the tightness of the binding and the specificity of the molecular interactions involved argue against this possibility. In any case, this is an issue that can be investigated by studying modern glycosidases. If heme binding is a functional ancestral feature (a product of selection), we may expect that at least some modern glycosidases show some inefficient, vestigial capability to bind heme, in keeping with the general principle that features that become less functional undergo evolutionary degradation[47,48]. No mention of heme binding to modern family 1 glycosidases can be found in the CAZypedia resource[22], but, of course, there is no reason why researchers in the glycosidase community should have tested heme-binding capabilities. As such, we have done so in this work for the four modern enzymes we already characterized in terms of catalysis (Fig. 4), i.e., the modern family 1 glycosidases from *Halothermothrix orenii*, *Thermotoga maritima*, *Marinomonas sp.* (strain MWL1), and *Saccharophagus degradans* (strain 2-40). When 5-aminolevulinic acid, the metabolic precursor of heme, was added to the culture medium, the four modern proteins were isolated with an appreciable amount of bound heme, although their heme-binding capability is clearly much reduced when compared with the ancestral glycosidase (Fig. 8 and Fig. S20). We furthermore carried out the same type of experiment with proteins corresponding to reconstructions at five nodes in the line of descent that leads from the ancestral glycosidase at node 72 to the modern glycosidase from *Halothermothrix orenii* (Fig. 8). We also found appreciable, but typically much lower amounts of bound heme, as compared with the "older" ancestral protein at node 72 (Fig. 8 and Fig. S21). Finally, we purified the heme-free forms of the proteins at these five ancestral nodes and studied their heme-binding capability in vitro. These proteins have glycosidase activity levels intermediate between those of the ancestral glycosidase at node 72 and the modern glycosidase from *Halothermothrix orenii* (Fig. S22) and unambiguously display in vitro heme-binding capability at micromolar concentrations, as shown by the presence of the Soret band in UV–VIS spectra (Fig. S23) and further confirmed by mass spectrometry (Figs. S24–S26). Evolutionary degradation of ancestral heme binding, however, is clearly revealed by analyses of elution profiles from gel filtration chromatography in terms of protein concentration, heme concentration, and glycosidase

activity (Fig. 8). Thus, while heme binding to the most ancient studied node (our ancestral glycosidase at node 72) produces active monomers to a large extent, a trend towards a decreased amount of heme-bound monomers and appearance of higher association states upon heme binding is observed in the evolutionary line leading to the modern glycosidase from *Halothermothrix orenii*. One interesting possibility is that heme binding to the monomers of the less ancient proteins brings about conformational changes that trigger protein association.

It emerges that heme binding to the ancestral glycosidase at node 72 is not an oddity or an artifact of reconstruction. In contrast, it appears probable that heme binding to ancient family 1 glycosidases did specifically occur, and that it also underwent degradation at an early evolutionary stage to lead to a rudimentary capability with substantial variability, as it is commonly observed with vestigial features that are not subject to selection[47]. Overall, this suggests a complex evolutionary history for this family of enzymes involving perhaps a fortuitous (i.e., contingent) early fusion event with a heme-containing domain. In this scenario, heme had a functional role in the isolated heme-containing domain, which was no longer required when the domain was fused with the larger glycosidase scaffold, thus enabling the subsequent degradation of the heme-binding capability. In order to find some evidence for this hypothesis, we used the Dali sever[49] to search in the Protein Data Bank for structural alignments of the alpha helices involved in heme binding from our ancestral glycosidase. However, we did not find any convincing match, as the best obtained structural alignments had RMSD values of ~4 Å and Z scores of 2 or higher (see Fig. S27 for further details). It is possible that the structure of the ancestral heme-containing domain was distorted upon fusion and subsequent evolution, and is therefore difficult to identify in searches of modern protein structures. Another possibility is that there was never a fusion event with a heme-containing domain and that heme was already present even at the most ancient stages in the origins of family 1 glycosidases. This would be consistent with an interpretation of cofactors as molecular fossils that facilitated the primitive emergence of proteins by selecting them from a random pool of polypeptides[50].

It is important to note at this stage that relevant protein-engineering implications are independent of any evolutionary interpretations. In this context, heme-binding to the ancestral glycosidases, regardless of its evolutionary origin and implications, opens up new engineering possibilities. This is so because directed laboratory evolution can be used to enhance or modify any functionality, provided that a certain level of functionality is used to start the process. The capability to "seed" levels of new functionalities may become a critical bottleneck in protein-engineering projects. Our work uncovers a heme-binding capability and a possibility of allosteric regulation that were previously unknown in glycosidase enzymes. Potential practical implications are briefly discussed below.

Metalloporphyrins are essential parts of many natural enzymes involved in redox and rearrangement catalysis and can be engineered for the catalysis of non-natural reactions[51]. Remarkably, however, the combination of the highly evolvable TIM-barrel scaffold and the catalytically versatile metalloporphyrins is exceedingly rare among known modern proteins. A porphyrin ring is found in only 13 out of the 7637 PDB entries that are assigned the TIM-barrel fold according to the CATH classification[52]. These 13 entries correspond to just two proteins. One of them, uroporphyrinogen decarboxylase, is an enzyme involved in heme biosynthesis, while in the other identified case, flavocytochrome B2, the bound heme is far from the active site. By contrast, heme appears at about 8 Å from the catalytic Glu171 in our ancestral glycosidase. While its connection with the

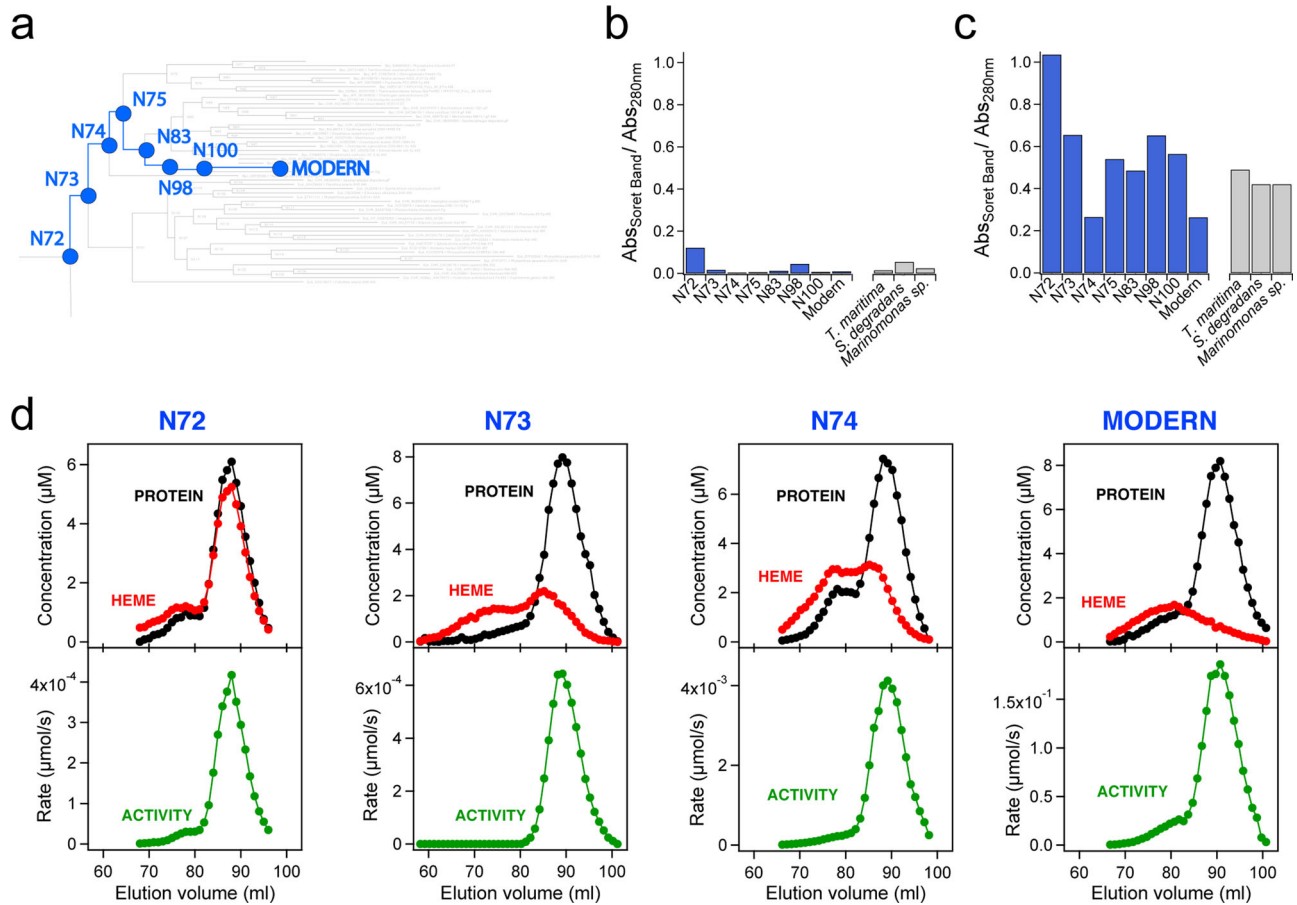

**Fig. 8 Evolutionary degradation of ancestral heme binding. a** Section of the phylogenetic tree used for the Bayesian analysis of family 1 glycosidases. The nodes in the evolutionary trajectory from node 72 to *Halothermothrix orenii* (labeled "MODERN") are highlighted and labeled. See Fig. S1 for a complete and detailed version of the phylogenetic tree. **b**, **c** Ratio of absorbance (Abs) at the maximum of the heme Soret band to the absorbance at the maximum of the protein aromatic absorption band for ancestral (see panel A) and modern family 1 glycosidases. Data in (**b**) correspond to protein preparations in which 0.4 mM 5-aminolevulinic acid (the metabolic precursor of heme) was added to the culture medium and the protein was purified by Ni-NTA affinity chromatography and further passage through a PD10 column (see Figs. S20 and S21 for further detail). For the data in (**c**), heme-free protein samples at ~5 µM were incubated for 1 h at pH 7 with a 5-fold excess of heme and free heme was removed through size exclusion chromatography (2 passages through PD10 columns) before recording the UV–VIS spectra (Fig. S23). **d** Profiles of protein concentration (bicinchoninic acid method[33]), heme concentration (pyridine hemochrome spectrum[3]) and glycosidase activity (with 0.25 mM 4-nitrophenyl-β-D-glucopyranoside) upon elution from gel filtration chromatography (HiLoad 16/600 Superdex 200 pg GE Healthcare). For these experiments, heme was gradually added to ~30 µM samples of ancestral (N72, N73, and N74) and modern (*Halothermothrix orenii*) glycosidases up to a ~5-fold excess and, after several hours, free heme was removed using PD10 columns. The elution volume for the main protein concentration peak is consistent with the monomeric association state (Fig. S3).

natural active site is blocked by several side chains in the determined structure (Fig. S17), it would appear feasible to use protein engineering to establish a conduit. As a simple illustration of this possibility, mutating Pro172, Asn173, Ile224, Leu226, Asn227, and Pro 272 to alanine in silico (Fig. S17 and Table S8) increase the accessible surface area of heme from barely 43 Å² to about 300 Å² and exposes the side of the heme facing the active site. The possibility thus arises that the engineering of metalloporphyrin catalysis, through rational design and/or laboratory evolution, would benefit from the evolutionary possibilities afforded by the flexible βα loops at a TIM-barrel catalytic face.

More immediate engineering possibilities arise from the allosteric modulation of catalysis in the ancestral protein, a phenomenon that, to our knowledge, has never been reported for modern glycosidases. Heme binding rigidifies the ancestral glycosidase and causes a several-fold activity enhancement. Heme is not expected to catalyze glycoside hydrolysis and, in

any case, the bound heme does not have access to the active site in the experimentally determined ancestral structure. The activity enhancement upon heme binding is therefore an allosteric effect likely linked to dynamics, as it is the case with other allosteric effects reported in the literature[53]. Regardless of whether this feature is truly ancestral or just a byproduct of the enhanced conformational flexibility of the putative ancestral glycosidase, it is clear that it can provide a basis for biosensor engineering. For instance, computational design and laboratory directed evolution could be used to repurpose the heme-binding site for the binding of a targeted substance of interest and to achieve a large concomitant change in glycosidase activity. The development of this application should be facilitated by the availability of a wide diversity of synthetic chemical probes for the sensitive detection of glycosidase activity[17]. In total, we anticipate that unusual combinations of protein features will generate new possibilities in protein biotechnology and engineering.

## Methods

**Ancestral sequence reconstruction.** Characterized GH1 protein sequences were retrieved from the Carbohydrate-active enzyme database (CAZy)[18], including β-glucosidase (accession numbers: ACI19973.1 and AAL80566.1), 6-P-β-glucosidase (AIY91871.1), β-mannosidase (AAL81332.1), and myrosinase (AAK32833.1). These characterized protein sequences were utilized as seeds to identify additional homologous sequences. GH1 homologs were retrieved for all three domains of life from GenBank (http://www.ncbi.nlm.nih.gov/) using BLASTp, with the cutoff threshold of $<1 \times 10^{-5}$. Sequences with the minimum length of 300 amino acids were included in the dataset. Taxonomically redundant sequences were excluded. A total number of 150 sequences were collected for further analysis.

Sequences were aligned using T-Coffee. Initial non-bootstrapped phylogenetic trees were constructed using RAxML (ver. 8.2.11) to identify major clusters and to eliminate spurious sequences[54]. The RAxML analysis was performed with the hill-climbing mode using the gamma substitution model. These initial trees were used as starting point for more thorough Bayesian analysis. MrBayes (ver. 3.2.6) was conducted using the WAG amino acid replacement model with a gamma distribution and invariable sites model for at least 1,000,000 generations, with sampling at intervals of 100 generations, and two runs with four chains per run in order to monitor convergence[55]. Twenty-five percent of sampled points were discarded as burn-in. The tree topology was broadly identical between the RAxML and MrBayes analyses.

Ancestral sequences were reconstructed using FastML (ver. 3.1) with the WAG amino acid replacement model with a gamma distribution for variable replacement rates across sites[56].

**Database searches.** We searched the CAZy database in order to ascertain the presence of porphyrin rings in reported glycosidase structures. We systematically went through all 167 glycoside hydrolase families of the database in March 2020. For each family, we checked the structure section and we individually examined all the links provided to the protein data bank. Overall, we examined 5565 PDB files corresponding to 1435 different glycosidase enzymes. We did not find a single example of a reported structure with a bound porphyrin ring.

We also used the CAZy database as a starting point of an extensive literature search for optimum temperature values of family 1 glycosidases. We examined the section of characterized enzymes for family 1 glycosidases, the references included in the corresponding GenBank links, as well as the publications that cite those references in a Google Scholar search. Several hundred published articles were examined for experimental activity versus temperature profiles and reported values of the optimum temperature. We found such data for 126 different family 1 glycosidases. In many cases, the oligomerization state of the enzymes was also provided in the original references. The environmental temperatures (optimum growth temperatures) of the corresponding host organisms could be found in most cases in the "Bergey's Manual of Systematic Bacteriology" although, in some cases, literature searches were performed to find the optimum temperatures. Most organisms were classified as hyperthermophilic, extreme thermophilic, thermophilic, mesophilic or psychrophilic in Bergey's manual or the relevant literature references. We have used this classification to color code Fig. 1c, since it leads to clear and intuitive data clusters. All the information related to the values of the optimum temperature for activity and the organismal living temperature is collected in Supplementary Dataset 1.

In order to find examples of proteins with the TIM-barrel fold and a bound porphyrin ring in the reported structures, we checked (March-2020) all entries in the protein data bank that are classified as TIM-barrels in the CATH database. We examined a total of 7637 PDB files and found a porphyrin ring in only 13 of them. These 13 structures correspond to two proteins: flavocytochrome B2, a multi-domain protein in which the porphyrin ring is located in the non-TIM-barrel domain, and uroporphyrinogen decarboxylase, which is an enzyme involved in heme biosynthesis.

**Protein expression and purification.** The different proteins studied in this work were purified following standard procedures. Briefly, genes for the His-tagged proteins in a pET24b(+) vector with kanamycin resistance were cloned into E. coli BL21 (DE3) cells, and the proteins were purified by Ni-NTA affinity chromatography in HEPES buffer. The His tag was placed at the C-terminus, i.e., at a position that is well removed from the catalytic face of the barrel, the regions of enhanced conformational flexibility in the ancestral protein and the heme-binding site. Since the ancestral protein is susceptible to proteolysis, we included protease inhibitors in all steps of the purification (cOmplete® EDTA-free Protease Inhibitor Cocktail from Roche, ref. 11873580001). Protein solutions were prepared by exhaustive dialysis against the desired buffer (typically 50 mM HEPES pH 7) or by passage through PD10 columns. Protein purity was assessed by gel electrophoresis (Fig. S28). Proteins were properly folded, as judged by circular dichroism spectra (Fig. S29) [Far-UV CD spectra from 250 to 210 nm were recorded for extant and ancestral glycosidases, at 25 °C, using a Jasco J-715 spectropolarimeter equipped with a PTC-348WI. Buffer conditions were 50 mM HEPES, pH 7.0, protein concentration was within 0.2–0.6 mg/mL range and a 1 mm pathlength cuvette was used. An average of 30 scans was performed in each case. Blank subtraction was always carried out prior to mean residue ellipticity calculation, $[\Theta]_{MRW}$].

**Analytical ultracentrifugation.** Samples of the ancestral glycosidase in HEPES 50 mM, NaCl 150 mM, pH 7.0 were used. The assays were performed at 48,000 rpm (185,463 xg) in an XL-I analytical ultracentrifuge (Beckman-Coulter Inc.) equipped with both UV–VIS absorbance and Raleigh interference detection systems, using an An-50Ti rotor. Sedimentation profiles were recorded simultaneously by Raleigh interference and absorbance at 280 nm. Differential sedimentation coefficient distributions were calculated by least-squares boundary modeling of sedimentation velocity data using the continuous distribution c(s) Lamm equation model as implemented by SEDFIT 16.1c[57]. These experimental values were corrected to standard conditions using the program SEDNTERP[58] (version 20120111 Beta) to obtain the corresponding standard values (s20,w).

Sedimentation equilibrium assays (SE) for GH1-N72 were carried out at speeds ranging from 8,000 to 11,000 rpm (5,152 xg to 9,740 xg) and at 280 nm, using the same experimental conditions and instrument as in the SV experiments. A last high-speed run (48,000 rpm, 185,463 xg) was done to deplete protein from the meniscus region to obtain the corresponding baseline offsets. Weight-average buoyant MW of GH1-N72 were obtained by fitting a single-species model to the experimental data using the HeteroAnalysis 1.1.60 program[59] once corrected for temperature and solvent composition with the program SEDNTERP[58] (version 20120111 Beta).

**Preparation of the ancestral protein with bound heme and determination of the heme to protein ratio.** Stock solutions of hemin (heme with iron in the +3 oxidation state) were prepared daily in 1.4 M sodium hydroxide. Prior to use, the stock solution was diluted (typically 1:100) into HEPES buffer 50 mM, pH 7 and this solution was immediately used.

The ancestral protein with bound heme was prepared by incubating the protein with a 5-fold excess of heme for about one hour, followed by passage through a PD10 column and a Superdex-200 column to eliminate the non-bound heme. The heme to protein ratio in the resulting protein samples could be roughly estimated from the absorbance of the Soret band and the protein band at 280 nm in UV–VIS spectra. This procedure is not exact because the Soret band may depend on the interactions of the bound heme and, also, heme can show some absorption at 280 nm. For more accurate characterization protein concentration was determined by the bicinchoninic acid method[33] with the Pierce™ BCA Protein Assay Kit (ThermoFisher Scientific). A method based on pyridine hemochrome spectra[34] was used to determine the amount of heme. Briefly, 25 μl of 0.1 M potassium ferricyanide were added to a mixture of 2 mL or pyridine, 2 mL 0.1 M NaOH and 2 mL water. This solution was mixed with the protein solution in a 1:1 volume ratio and an excess of sodium dithionite was added. Lastly, the amount of heme was calculated from the absorbance of the pyridine hemochrome at 556 nm after correction for the absorbance of a blank. Using this approach, a heme to protein stoichiometry of 1.03±0.03 was determined from 5 independent measurements.

**UPLC mass spectrometry.** Ultra performance liquid chromatography (UPLC) was performed using a Waters Acquity H Class UPLC connected to a mass spectrometry Waters Synapt G2 Triwave® system. A 2.1 × 100 mm Protein BEH C4 column of 300 Å pore size and 2.1 μm particle size at a flow of 0.2 mL/min was used for chromatography. The mobile phase was a mixture of 0.1% formic acid–water (A) with 0.1% formic acid–acetonitrile (B) and the elution gradient were as follows: 0–10.33 min, 98–55% A; 10.33–20 min, 55–30% A; 20–21.57 min, 30% A; 21.57–23,33 min, 30–2% A; 23.33–30 min, stay 2% A. Mass spectrometry conditions were as follows: the ionization source of ESI was operated inion mode of positive (ESI +) and 2.2 kV of capillary voltage. Temperature of desolvation was 400 °C, and ion source was 100 °C. Desolvent and cone gas (nitrogen) flow velocity were 600 L/h.

**Microscale thermophoresis quantification of heme-protein interaction.** The motion of molecules in microscopic temperature gradients (microscale thermophoresis) is sensitive to changes in properties induced by a binding event and can be used to quantify a diversity of intermolecular interactions[60]. For these experiments, we used a His-tagged protein labeled with a fluorescent probe using the His-tag labeling kit from NanoTemper technology. A 200 nanomolar protein solution was titrated at 25 °C with heme concentrations ranging from 55 nM to 2 μM. We did not use higher heme concentration to minimize the possibility of heme association, a process that would decrease the concentration of the monomeric heme that is competent for binding. The experiments were performed with Monolith NT.115 pico from NanoTemper technology. The data were acquired with MO. Control software, version 1.6 (NanoTemper Technologies GmbH). The binding curve and affinity were modeled and analyzed in the MO.Control software, version 1.6. Three replicate experiments were performed to yield an average value for the heme dissociation constant of 547 ± 110 nanomolar. Relevant plots and validation reports for the three experiments are shown in Fig. S14.

**Activity determinations.** Glucosidase and galactosidase activities were tested following the absorbance of p-nitrophenol at 405 nm upon the hydrolysis of 4-nitrophenyl-β-D-glucopyranoside and 4-nitrophenyl-β-D-galactopyranoside[27]. Rates were calculated from the initial absorbance vs. time slope and the known extinction coefficient of p-nitrophenol at pH 7. Experiments at different substrate

concentrations were carried out to arrive at Michaelis plots for the ancestral and several modern glycosidases studied in this work. For the rate determination at a wide range of substrate concentrations we used a protocol designed to minimize any changes in buffer composition that could distort the profiles. Thus, for a rate measurement at a given substrate concentration, an enzyme solution in HEPES buffer at pH 7 was mixed with an equal volume of substrate dissolved in pure water. To minimize pH changes, the initial enzyme solution was prepared in 200 mM buffer to yield a final buffer concentration of 100 mM. We confirmed that the pH changes upon mixing were negligible. See legends to Figs. S6–S9 for details on data analysis.

As it is common in the literature, values of the optimum activity temperatures were determined from the profiles of activity versus temperature derived from measurements performed after several-minute incubations at each temperature[25]. Briefly, the protein was incubated at the desired temperature with 1 mM substrate in HEPES buffer 50 mM pH 7 and, after 10 min, the reaction was stopped by adding sodium carbonate to a concentration of 0.5 M. The amount of substrate hydrolyzed was determined from the absorbance of p-nitrophenol at 405 nm. We confirmed that the 10-min incubation only hydrolyzed a fraction of the substrate present and, therefore, that the amount of substrate hydrolyzed after a 10-min. incubation is a suitable metric of enzyme activity. Profiles of activity versus temperature were determined using both 4-nitrophenyl-β-D-glucopyranoside and 4-nitrophenyl-β-D-galactopyranoside. The profiles for the ancestral glycosidase are shown in Fig. 1b and those for the modern glycosidase from *Halothermothrix orenii* are given in Fig. S5. In all cases, the profiles show a sharp maximum from which an unambiguous determination of the optimum temperature is possible. Note also that there is good agreement between the optimum temperature values derived using the two different substrates. Differential scanning calorimetry experiments were performed as we have previously described in detail[9].

Glycosidase substrates were obtained from commercial sources, except 4-nitrophenyl-β-D-glucopyranoside-6-phosphate, which was prepared by us on the basis of a published procedure[61]. The chemical identity of the prepared compound was confirmed by mass spectrometry and nuclear magnetic resonance.

**Proteolysis experiments.** For proteolysis experiments, the ancestral glycosidase and the modern glycosidase from *Halothermothrix orenii* at a concentration of 1 mg/mL were incubated at 25 °C with thermolysin for different times in HEPES buffer 50 mM pH 7 containing 10 mM calcium chloride. Stock solutions of thermolysin were prepared fresh in the same solvent at a concentration of 1 mg/mL and were diluted 1:10 when added to the protein solution. The reaction was stopped by the addition of EDTA to a final concentration of 12.5 mM and aliquots were loaded into 15% (w/v) SDS-PAGE gels for electrophoresis. In some experiments, fragments separated by electrophoresis were extracted, desalted and subjected to LC-MS/MS analysis for mass determination. Fragment masses were determined by MALDI and their sequences were investigated using peptide mapping fingerprinting and MALDI-TOF/TOF (Fig. S2). This allowed us to locate approximately the cleavage sites, as shown in Fig. 2c.

**Crystallization and structure determination.** The ancestral glycosidase, dissolved in 150 mM NaCl, 50 mM HEPES pH 7.0, was concentrated to 35 mg/mL and to 70 mg/mL for the vapor-diffusion (VD) and counter-diffusion crystallization experiments, respectively. We checked by SDS electrophoresis that the concentrated protein used for crystallization was not proteolyzed. Hanging-drops VD experiments were prepared by mixing 1 μL of protein solution with the reservoir, in a 1:1 ratio, and equilibrated against 500 μL of each precipitant cocktail HR-I (Crystal Screen 1, Hampton Research). Capillary counter-diffusion experiments were set up in capillaries of 0.3 mm inner diameter using the CSK-24, AS-49 and PEG448-49 screening kits[62]. A similar procedure was followed for the crystallization of the ancestral glycosidase-heme complex, using two fixed concentrations at 75 and 30 mg/ml for the counter-diffusion and VD experiments. Experiments were performed at 293 K.

Crystals of the ancestral glycosidase were obtained only in condition #41 of HR-I, whilst the GH1N72-Heme complex crystallized in conditions #6 and #9 of HR-I and PPP8 of the mix of PEG counter-diffusion screen. Crystals were extracted either from the capillary or fished directly from the drop and subsequently cryo-protected by equilibration with 15 % (v/v) glycerol prepared in the mother liquid, flash-cooled in liquid nitrogen and stored until data collection. Crystals were diffracted at the XALOC beamline of the Spanish synchrotron light radiation source (ALBA, Barcelona). Indexed data were scaled and reduced using the CCP4 program suite[63].

Initial data sets were obtained for the ancestral glycosidase crystals diffracting the X-ray to 2.5 Å. The clean (without water, ligands, etc.) 3D model of the β-glucosidase from *Thermotoga maritima* (PDB ID. 2J78) [https://www.rcsb.org/structure/2J78] was used as search model for molecular replacement[63]. Two monomers were found in the asymmetric unit as expected from the Matthews coefficient for the P2(1) space group. Refinement, including Titration-Libration-Screw (TLS) parametrization, water pick, and model validation was carried out with PHENIX suite[64]. Unidentifiable amino acids in the highly disordered region have been assigned as poly-UKN chains C and D corresponding to the 18 and 14 (poly-Alanine) of chains A and B, respectively.

Crystals of the ancestral glycosidase-heme complex belong to the same space group than the ancestral glycosidase but were not isomorphous. The determined unit cell was bigger accommodating three polypeptide chains in the asymmetric as determined from the Matthews coefficient. A similar protocol was followed to place the three monomers in the unit cell by molecular replacement and to refine the structure. After a first refinement round the presence of one protoporphyrin ring in each polypeptide chain was determined. It was also clear that disordered regions of the ancestral glycosidase model were visible in the heme complex model.

The summary of data collection, refinement statistics, and quality indicators are collected in Table S9. The coordinates and the experimental structure factors have been deposited in the Protein Data Bank with ID 6Z1M [https://www.rcsb.org/structure/6Z1M] and 6Z1H [https://www.rcsb.org/structure/6Z1H] for the ancestral glycosidase with and without bound heme, respectively.

Figures displaying 3D-structures have been prepared using PyMOL (The Pymol Molecular Graphics System, Schrödinger, LLC). The 2D-interaction diagram of Fig. 7a was prepared using LigPlot+ (https://www.ebi.ac.uk/thornton-srv/software/LIGPLOT/).

**Molecular dynamics simulations.** Molecular simulations were performed on both ancestral glycosidases (this work, PDB ID: 6Z1M [https://www.rcsb.org/structure/6Z1M], 6Z1H [https://www.rcsb.org/structure/6Z1H]) and the modern glycosidase from *Halothermothrix orenii* (PDB ID: 4PTV [https://www.rcsb.org/structure/4PTV]). The structure of the heme-free ancestral glycosidase was obtained by manually deleting the heme coordinates from the corresponding heme-bound crystal structure. The missing regions of the ancestral glycosidases were reconstructed using MODELLER. Histidine protonation states were selected based on empirical pK$_a$ estimates performed using PROPKA 3.1 and visual inspection. All other residues were placed in their standard protonation states at physiological pH. The heme group was described using a bonded model, creating a bond between the Tyr264 side chain and the Fe(III) atom of the heme (Fig. S30). We used MCPB.py as implemented in AMBER19[65] to obtain the necessary parameters for creating the bonding pattern between the Fe(III) atom of the heme and the 4 nitrogen atoms of the heme and the tyrosine side chain oxygen. The resulting structure was then optimized, and frequency calculations were performed at the ωB97X-D/6-31 G* level of theory followed by the Seminario method[66] to obtain the force constants from the Hessian of the frequency calculation. This functional is a long-range (LC) corrected hybrid functional (see Chai and Head-Gordon[67] and references cited therein), which describes short-range interactions using an exchange functional, and long-range interactions using 100% Hartree-Fock exchange. ωB97X-D further improves on the concept of LC functionals, by systematic optimization of the functional, including the inclusion of an extra parameter that allows for an adjustable fraction of short-range exchange. In addition, this functional incorporates an empirical dispersion correction, following the DFT-D scheme[68]. We chose this functional for our parameterization as it yielded optimized structures of the heme complex that were similar to that observed in the heme complex, which is important in order to be able to maintain the heme in the binding pocket in a planar (non-distorted) conformation in our bonded model. This was further corroborated in our MD simulations, where the heme maintained a planar conformation without any unphysical distortions of dihedral angles. We note that only the heme and the Tyr264 side chain were considered as QM atoms in our model, as this is the region that was necessary to parameterize in our bonded model, following Li and Merz[69].

Partial charges were obtained for the heme and for the Tyr264 side chain at the HF/6-31 G* level of theory, using the restrained electrostatic potential (RESP) approach, following the MCPB.py protocol, and performing the calculations using Gaussian 09 Rev. E.01. Periodic boundary conditions (PBC) were used, and all systems were solvated in a truncated octahedral box filled with TIP3P water molecules[70], with 10 Å from the solute to the box edges in all directions. The truncated octahedron can fill space without leaving any gaps, and since our protein has a globular shape, a truncated octahedral box is the most suitable box shape to reduce the number of water molecules necessary to fill the box, which saves substantial computational time. with a distance of 10 Å from the solute to the surface of the box. The box was then filled with TIP3P water molecules. Na+ and Cl− counter ions were added to the system to neutralize each enzyme. The protein was described using the AMBER ff14SB force field[71], and the heme was described using the General AMBER Force Field (GAFF)[72].

Following system preparation, the LEaP module of AMBER19 was used to generate the topology and coordinate files for the MD simulations, which were performed using the CUDA version of the PMEMD module of the AMBER19 simulation package. The solvated system was first subjected to a 5000 step steepest descent minimization, followed by a 5000 step conjugate gradient minimization with positional restraints on all heavy atoms of the solute, using a 5 kcal mol⁻¹ Å⁻² harmonic potential. The minimized system was then heated up to 300 K using the Berendsen thermostat[73], with a time constant of 1 ps for the coupling, and 5 kcal mol⁻¹ Å⁻² positional restraints (again a harmonic potential) applied during the heating process. The positional restraints were then gradually decreased to 1 kcal mol⁻¹ Å⁻² over five 500 ps steps of NPT equilibration, using the Berendsen thermostat and barostat to keep the system at 300 K and 1 atm. For the production runs, each system was subjected to either 500 ns of sampling in an NPT ensemble at constant temperature (300 K) and constant pressure (1 atm), controlled by the Langevin thermostat, with a collision

frequency of 2.0 ps$^{-1}$, and the Berendsen barostat with a coupling constant of 1.0 ps. A 2 fs time step was used for all simulations, and snapshots were saved from the simulation every 5 ps. The SHAKE algorithm[74] was applied to constrain all bonds involving hydrogen atoms. A 10 Å cutoff was applied to all nonbonded interactions, with the electrostatic interactions being treated with the particle mesh Ewald (PME) approach[75]. 10 independent simulations were performed for each starting structure during 500 ns (for RMSD convergence, see Fig. S18). All the subsequent analyses were performed with the CPPTRAJ toolkit from Ambertools19[65]. Parameters used to describe the heme, input files, snapshots from our simulations, and simulation trajectories (with water molecules and ions removed to save file size) are available for download from Zenodo (https://zenodo.org) at https://doi.org/10.5281/zenodo.3857791.

**Reporting summary**. Further information on research design is available in the Nature Research Reporting Summary linked to this article.

## Data availability
Data included in the figures and supporting the findings of this study are available from the corresponding authors upon reasonable request. Atomic coordinates and the experimental structure factors have been deposited in the Protein Data Bank (https://www.rcsb.org) with ID 6Z1M and 6Z1H for the ancestral glycosidase with and without bound heme, respectively. Parameters used to describe the heme, input files, snapshots from our molecular dynamics simulations, and simulation trajectories (with water molecules and ions removed to save file size) are available for download from Zenodo (https://zenodo.org) at https://doi.org/10.5281/zenodo.3857791. Source data are provided with this paper.

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

## Acknowledgements
This work was supported by Human Frontier Science Program Grant RGP0041 (J.M.S.-R., E.A.G., B.S., and S.C.L.K.), NIH grant R01AR069137 (E.A.G.), Department of Defense grant MURI W911NF-16-1-0372 (E.A.G.), the Swedish Research Council (2019-03499) (S.C.L.K.), the Knut and Alice Wallenberg Foundation (2018.0140 and 2019.0431) (S.C.L.K.), Spanish Ministry of Economy and Competitiveness/FEDER Funds Grants BIO2015-66426-R (J.M.S.-R.) RTI2018-097142-B-100 (J.M.S.-R.) and BIO2016-74875-P (J.A.G.). The simulations were enabled by resources provided by the Swedish National Infrastructure for Computing (SNIC) at UPPMAX partially funded by the Swedish Research Council through grant agreement no. 2016-07213. We acknowledge the Spanish Synchrotron Radiation Facility (ALBA, Barcelona) for the provision of synchrotron radiation facilities and the staff at XALOC beamline for their invaluable support. We are also grateful to Victoria Longobardo Polanco (Proteomic Unit, Institute of Parasitology and Biomedicine "López-Neyra") for help with mass spectrometry experiments and data analyses and to Juan Román Luque Ortega (Molecular Interactions Facility, Centro de Investigaciones Biológicas Margarita Salas) for help with ultra-centrifugation experiments and data analyses.

## Author contributions
B.S., S.C.L.K., E.A.G., and J.M.S.-R. designed the research. G.G.-A. and L.I.G.-R. prepared the protein variants and designed, performed and analyzed experiments addressed at determining their catalytic and biophysical features, under the supervision of V.A.-R. and B.I.-M., who also provided essential input regarding the interpretation of these properties. V.A.R. was in charge of mass spectrometry, ultracentrifugation, and thermophoresis experiments. Y.H. carried out ancestral sequence reconstruction under the supervision of E.A.G., who also provided essential input for the interpretation of the results in an evolutionary context. D.P. performed homology modeling under the supervision of S.C.L.K. Organic synthesis was performed by J.J. and J.M.C. who provided essential input regarding the properties of the synthesized compound. A.R.-R. carried out MD simulations under the supervision of S.C.L.K., and they provided the general interpretation and implications of the simulations. L.I.G.-R. and V.A.R. carried out protein crystallization. J.A.G. determined the X-ray structures and provided essential input regarding their interpretation and implications. J.M.S.-R. wrote the first draft of the manuscript to which B.S., S.C.L.K., and E.A.G. added crucial paragraphs and sections. All authors discussed the manuscript, suggested modifications and improvements, and contributed to the final version.

## Funding

## Competing interests
The authors declare no competing interests.
