## [Peer Review File · Nature Communications]

REVIEWER COMMENTS

Reviewer #1 (Remarks to the Author):

Thanks for a really interesting paper.

Lines 107-108:

You need to mention that the TIM barrel is a superfold in the sense of Orengo CA, Jones DT, Thornton JM, *Nature*, 372, 631-4 (1994). Thus, sharing of this particular fold does not necessarily imply evolutionary relatedness. There's a detailed discussion of this in reference 12 of the current manuscript, Nagano et al. (2002).

Line 133: You should say ...

family 1 glycosidase (GH1) protein sequences

(not glycosidases, since the last s is not used in the usual English idiom)

Lines 253-254:

This kind of inefficiency often seems to be observed with reconstructed ancient enzymes (sometimes along with broader substrate preferences). Is this because the historical enzyme was less efficient or because the best-guess reconstructed version still differs substantially from the actual sequence of the real-life ancestor?

Line 997

Please say more about this wB97X-D calculation, which I think uses a somewhat unusual range-separated functional. Could you explain a little more about this level of theory? Is the whole enzyme in the QM system here, or just certain atoms? More details would be appreciated.

Reviewer #2 (Remarks to the Author):

"Novel heme-binding enables allosteric modulation in an ancient TIM-barrel glycosidase"
Gamiz-Arco et al

The authors report the resurrection and characterization of a hypothetical family-1 glycosidase from the common ancestor of bacteria and eukaryotes. The ancestral enzyme is a ($\beta\alpha$)₈ barrel, like extant family-1 glycosidases, and binds heme in the manner of an allosteric activator. This is a novel finding, as the literature has no reports of extant glycosidases that bind porphyrin. The ancestral enzyme is active with standard substrates, though with catalytic efficiency $\sim 1000\times$ lower than extant enzymes. In the presence of heme (which binds distinct to the active site), the ancestral enzyme becomes more rigid and displays increased turnover rates with both pNP-Glu and pNP-Gal (although K_M values also increase). The authors postulate that the ancestor displays properties of a generalist enzyme under selection for more efficient, specific catalysis; they further speculate that the heme-binding domain may be the result of a happy gene fusion accident.

The authors have demonstrated highly competent characterization of their ancestral enzyme and teased out a likely role for heme. However, to reach the caliber of *Nature Communications*, I would like to see further investigation of the changing role of heme in these enzymes over evolutionary time. As it stands, the glycosidase ancestor represents an outlier with little context for how it could have arisen and disappeared to. I recommend that the authors reconstruct more recent nodes in their tree

to trace how affinity for heme – and its effect on catalysis and/or protein dynamics – may have diminished over time. This would also corroborate the single result. The authors speculate that extant glycosidases may have vestigial heme-binding characteristics, but did they assay any for binding? Alternatively, the authors could generate simple error-prone libraries of the ancestral enzyme to determine how easily the role of heme could be replaced by side chain dynamics. In a reciprocal experiment, extant glycosidases could be randomized to determine whether they could bind heme. The authors did not elaborate on how the heme-binding motif in the ancestor differs in extant enzymes.

I would have appreciated a more in depth discussion on heme's role in catalysis.

I have listed further points below:

- S3 *Thermotoga maritima* misspelled
- F3 needs key for color coding
- F4 and Table S3 and all assay figures – please list replicates (i.e. biological and technical – I could not find this information in the methods either)
- F4 – include another graph of k_{cat}/K_M s. It is useful to see how turnover rate and Michaelis constants differ between the enzymes, but selective pressure acts at the level of the organism and therefore metabolic flux --- of which k_{cat}/K_M is a large component -- is more important. Showing just these two graphs ablates the fact that the ancestral glycosidase has essentially the same k_{cat}/K_M for pNP-glu in the presence and absence of heme. The increase in k_{cat} and K_M cancel each other out.
- The text mentions the proficiency of glycosidases. Please include uncatalysed rate and proficiency for each activity. This is particularly useful when discussing the poor activities of ancestral enzymes.
- S5 comment on poor saturation of enzymes with pNP-gal – were higher concentrations assayed or were K_M values extrapolated from the data? What is the limit for pNP-gal solubility in the assay?
- The paper describes the ancestral enzyme as a generalist because it demonstrates mM-range K_M s for both pNP-glu and pNP-gal. However, the ancestor has a k_{cat}/K_M for pNP-glu that is 6-fold higher than that for pNP-gal. *H. orenii* has a 7-fold higher k_{cat}/K_M for pNP-glu vs pNP-gal. Is the ancestor really a generalist? Why aren't the extant enzymes more specialized?
- If specialization occurs at the level of K_M , then one could argue that the ancestor does not have physiologically-relevant activity for either substrate especially in the presence of heme, which increases the K_M for both substrates.
- Did the authors assay the ancestor with more disparate substrates?
- Are the residues that interact with heme conserved in the extant proteins?
- Fig 6 has too many panels. Include A, B and a version of D or E.
- How many types of glycosidases are predicted to have been present in LUCA?

Reviewer #3 (Remarks to the Author):

This manuscript describes a very intriguing result from ancestral sequence reconstruction whereby the process of reconstruction has "discovered" a heme binding activity for an ancient GH1 family glycosidase enzyme. The authors describe a range of biochemical experiments to demonstrate the stoichiometric binding of heme and the relationship between binding and enzymatic activity. Molecular dynamics simulations are additionally described to model the dynamics of the ancestral structures (as determined by crystallography).

The conundrum for the reader is whether this is an interesting oddity or whether heme binding is a significant evolutionary intermediate on the trajectory (over deep time) for GH1 TIM barrel glycosidase enzymes. It appears that heme binding is robust and stoichiometric. The fact that heme binding improves k_{cat} (although not k_{cat}/K_M) is further strong evidence for this observation being significant. However, some of the assertions made by the authors are rather weak and deserve further analysis. In my opinion, major revision may provide further evidence that strengthens the case for this being a

significant evolutionary finding.

Major points:

The structure for the apo-ancestral enzyme has two loops missing due to disorder. These loops become (mostly) ordered upon heme binding. The authors rightly state that no extant GH1 enzymes display heme binding. Thus, the observation of a heme binding site is surprising and very interesting. They further speculate that a heme binding module may have been grafted onto the TIM barrel during evolution. This raises several questions:

1. The statement in Line 395 "flexibility is greatly enhanced and extends to large parts of the structure" is reiterated in several places in the manuscript. However, is it not more accurate to say that there are two flexible loops in the structure and that the effect is localised to these loops?
2. From a cursory look at the PDB, the extant thermophilic homologue from *Halothermothrix orenii* that is used as a comparison throughout the manuscript has a large PEG molecule bound to one of these "flexible" loops. Other closely related structures have e.g. hydrophobic groups bound in this region (see 2WC4 with a substrate analogue). Thus, it may be the case that the loops in question are "binding loops" in a general sense. A thorough review of the GH1 family enzymes will reveal this. Further, many GH enzymes have auxilliary domains. Are these loops common locations for the insertion of auxilliary domains?
3. Related to 2. the authors state on line 437: "Albeit limited, our results suggest that some modern family 1 glycosidases may retain a vestigial capability to bind heme." This is a very intriguing claim that deserves further analysis (see 4. below).
4. If the heme binding module has been grafted onto the TIM barrel, then it should be possible to find evidence for this hypothesis. For example, the authors state that an early fusion event with a heme-containing domain is a possibility. This can be tested. Is there any evidence for a module from the heme binding proteins in the PDB? There are numerous approaches to identify modular components of structures related to binding to test this assertion.
5. The molecular dynamics experiments deserve a greater profile in the manuscript in my opinion. They appear to indicate some very interesting trends (Figure S9). For example, the starting structures for the ancestral sequences appear to be a long way from the equilibrium structure. What is the relationship between the starting structure (presumably the crystal structure) and the equilibrium structures for the dynamics trajectories? It is interesting that the contemporary crystal structure for *Halothermothrix orenii* is much closer to the equilibrium MD ensemble. The standard deviation for the RMSD for the heme-bound structure is greater than that for the non-heme bound structure. What is the reason for this (somewhat counterintuitive) result? Does this shed light on the relationship between heme binding and catalysis?
6. Figure 2D is not terribly informative without reference to table S4. Is it possible to label the residues in figure 2D?

The enzymatic assays are well done, however the analyses need some revision in my opinion. The authors cite Wolfenden and state that glycosidases accelerate the chemical reaction by up to 17 orders of magnitude. In light of this, a value for k_{cat} of $\sim 0.1 \text{ s}^{-1}$ still represents a rate enhancement of ~ 15 orders of magnitude. Hence, considering the statement "the comparatively low activity of the ancestral protein is likely linked to its conformational flexibility" raises some questions given that the differences in rate are very subtle when compared to the rate enhancement which remains extremely high in both cases (i.e for both gluco and galacto substrates). There is no evidence that the conformational flexibility is influencing the very subtle differences in rate. Can the authors point to some evidence for this?

The kinetics for 4-PNP-glucopyranoside are intriguing. k_{cat} differs by a factor of 3.4 whereas k_{cat}/K_m values are approximately equal when apo- and heme-bound enzymes are compared. Thus, heme binding is significantly impairing K_m . Indeed, a K_m value of $\sim 9 \text{ mM}$ warrants some discussion. This is approaching a value commensurate with non-specific binding. This suggests to me that glucose (and galactose) are not the preferred substrates for this ancestral enzyme. Have the authors searched for alternative substrates? It is certain that the natural substrates are not 4-PNP-

gluco(galacto)pyranosides.

Figure 5C needs some explanation. What is the curve fitted to the 1 microM data? Why has the curve not been simultaneously fitted to the 100 nM and 50 nM data? The caption states that heme "binding is strong". What is the binding constant for heme?

Citing a paper from 1976 to justify the statement that ancient enzymes are generalists is not convincing.

In summary, this manuscript describes a very interesting and potentially significant finding - heme binding in a reconstructed GH1 enzyme. However, in my opinion, there are several hypotheses and speculative statements presented with relatively weak evidence or without evidence. This can potentially be rectified by further analyses.

Minor points:

Figure 2. The homology model in Figure 2A can be removed as it does not add anything to the analyses.

Line 383 "The TIM-barrel is the most common protein fold" - I don't think that this is the case. Perhaps the most common enzyme fold?

Table S6 - Rmerge for ancestral-heme is incorrect.

Figure 3 caption... "[Adrian describe clustering algorithm...]"?

Reviewer #4 (Remarks to the Author):

This manuscript provides an interesting account of an ancestral sequence reconstruction (ASR) experiment that has yielded a glycoside hydrolase (GH) with the curious ability to bind heme. This appears to be the first report of a heme-binding GH. Heme-binding increased this GH's activity (k_{cat}/K_m) by a factor of three.

It is proposed that this heme-binding ASR-generated GH may approximate an early stage in the evolution of the GH1 family and that this sequence may represent a useful chassis for the directed evolution of GHs with novel activities or allosteric activation mechanisms. However, these ideas are not explored in this preliminary investigation.

While the heme-binding of this GH is intriguing, I have two main reservations about the significance of this work:

1) It remains unclear if heme-binding is a characteristic shared by many ASR-generated sequences from different nodes of the GH1 phylogenetic tree, or if this is merely an artefact observed for this particular sequence. Having established a structural basis for heme binding, the authors could easily perform sequence alignments with ASR sequences from other nodes and even modern sequences to determine if the heme-binding motif is conserved. If so, it would be good to express and characterise these putative heme-binding proteins too. That would prove that this phenomenon is unlikely to be an artefact. The authors did initially explore three ASR sequences (nodes 72, 73 and 125) before abandoning N73 and N125 due to their propensity to aggregate... it is unclear if they ever tried rescuing these proteins by the addition of heme. More work is needed to establish the significance of heme-binding by this single ASR protein.

2) The activation upon heme binding is very modest – a factor of three. While this subtle effect is claimed to be useful as a starting point for biosensor development, this idea remains an unrealised aspiration. Furthermore, it remains unclear how a glucosidase activity might be used as a readout for heme concentrations.

A less significant issue I had with the manuscript was that I found the narrative to be somewhat disjointed. The heme-binding nature of this protein appeared very early on with the structural work without being addressed until much later. The authors could perhaps rethink how some of these figures and results are assembled to provide a smoother transition between topics.

In addition to the above, a few suggestions to improve this manuscript include:

- 1) Figure S3. SEC with external calibration is not a reliable way to determine protein oligomerisation state. Please perform SEC-MALS, AUC or SAXS to determine oligomerisation state in solution.
- 2) Figure 2A. Why include a homology model when you have the actual structure? This doesn't add any value to the work.
- 3) Figure 2C. Proteolysis is a crude method of assessing a protein's structural dynamics, since increased proteolysis is both fold and sequence related. Please consider using NMR or HDX experiments to more reliably address the question of protein dynamics.
- 4) The authors should expand their assessment of the enzyme's substrate preference beyond simple synthetic PNP-glycosides to include biologically relevant substrates like cellobiose, laminaribiose, gentiobiose and lactose. Since the N72 node also includes enzymes active on Glc-6-P, the authors should also assess this ASR enzyme for activity on this PNP-Glc6P. I appreciate that it is not commercially available but it is easy enough to prepare using hexose kinase.
- 5) It would be good to see some MS data for heme from the purified protein to support the structural assignment. Also, no attempt appears to have been made to measure the affinity of heme for this protein. This should be rectified.
- 6) Figure 6C-E was all a little unclear to me. Some or all of these images need to be enlarged, better annotated and moved to the SI. Figure 6E – in silico mutation work – is highly conjectural and is probably best removed altogether.

Minor points

Line 207-208. Is 2.5Å 'good' resolution?

Line 776. 50 nM not 50 nm.

To summarise, this manuscript reports a very interesting observation but it fails to demonstrate if this represents an important step in the evolution of the GH1 family or just a curious artefact of ASR. More rigorous experiments are required to support the authors claims around protein dynamics, substrate specificity and affinity for heme.

Reviewer #1 (Remarks to the Author):

Thanks for a really interesting paper.

Author response: We are delighted that the reviewer finds our work of interest.

Lines 107-108:

You need to mention that the TIM barrel is a superfold in the sense of Orengo CA, Jones DT, Thornton JM, Nature, 372, 631-4 (1994). Thus, sharing of this particular fold does not necessarily imply evolutionary relatedness. There's a detailed discussion of this in reference 12 of the current manuscript, Nagano et al. (2002).

Author response: This is certainly an important distinction and we thank the reviewer for highlighting this. In the revised version, we mention (see text highlighted on page 3) that common ancestry between different TIM-barrel sequence families cannot be unambiguously demonstrated and that, therefore, the TIM-barrel can be viewed as a superfold in the sense of Orengo et al., 1994.

Line 133: You should say ...

family 1 glycosidase (GH1) protein sequences (not glycosidases, since the last s is not used in the usual English idiom).

Author response: Thanks for noting this. It has been corrected in the revised version.

Lines 253-254:

This kind of inefficiency often seems to be observed with reconstructed ancient enzymes (sometimes along with broader substrate preferences). Is this because the historical enzyme was less efficient or because the best-guess reconstructed version still differs substantially from the actual sequence of the real-life ancestor?

Author response: We have shown that ancestral sequence reconstruction is remarkably accurate for the vast majority of inferred position (see, for example, Randall et al., Nature Communications, 7_12847, 2016). Still, as we note in the manuscript (page 11) the critical active site residues in our reconstructed ancestral glycosidase match those in modern highly active glycosidases. Therefore, lower activity of the ancestral protein is probably not due to any gross misconstruction at the active site. Furthermore, as we also note in the revised version (page 7), the ancestral enzyme is not really inefficient per se, since its turnover number is about 13 orders of magnitude above the rate of the uncatalyzed reaction. It is simply not as efficient as some modern glycosidases, a fact that appears to be linked to enhanced conformational flexibility.

Line 997

Please say more about this wB97X-D calculation, which I think uses a somewhat unusual range-separated functional. Could you explain a little more about this level of theory? Is the whole enzyme in the QM system here, or just certain atoms? More details would be appreciated.

Author response: wB97X-D is an optimized long-range corrected (LC) hybrid functional, where the exchange parameters have been systematically optimized, and where an additional parameter is introduced to allow for an adjustable fraction of short-range exchange (wB97X), improving the performance of the functional (see discussion Chai and Head-Gordon, PCCP 10 (2008), 6615). In addition, wB97X-D incorporates a DFT-D type empirical dispersion correction which improves its treatment of non-covalent interactions compared to wB97X. Only the heme and tyrosine side chain were described in the QM system for the parameterization, as now described in the Methodology section on page 34.

Reviewer #2 (Remarks to the Author):

“Novel heme-binding enables allosteric modulation in an ancient TIM-barrel glycosidase” Gamiz-Arco et al

The authors report the resurrection and characterization of a hypothetical family-1 glycosidase from the common ancestor of bacteria and eukaryotes. The ancestral enzyme is a ($\beta\alpha$)₈ barrel, like extant family-1 glycosidases, and binds heme in the bacteria and eukaryotes. The ancestral enzyme is a ($\beta\alpha$)₈ barrel, like extant family-1 glycosidases, and binds heme in the manner of an allosteric activator. This is a novel finding, as the literature has no reports of extant glycosidases that bind porphyrin. The ancestral enzyme is active with standard substrates, though with catalytic efficiency ~1000x lower than extant enzymes. In the presence of heme (which binds distinct to the active site), the ancestral enzyme becomes more rigid and displays increased turnover rates with both pNP-Glu and pNP-Gal (although KM values also increase). The authors postulate that the ancestor displays properties of a generalist enzyme under selection for more efficient, specific catalysis; they further speculate that the heme-binding domain may be the result of a happy gene fusion accident.

The authors have demonstrated highly competent characterization of their ancestral enzyme and teased out a likely role for heme.

Author response: We are delighted that the reviewer finds that our work has been competently carried out.

However, to reach the caliber of Nature Communications, I would like to see further investigation of the changing role of heme in these enzymes over evolutionary time. As it stands, the glycosidase ancestor represents an outlier with little context for how it could have arisen and disappeared to. I recommend that the authors reconstruct more recent nodes in their tree to trace how affinity for heme – and its effect on catalysis and/or protein dynamics – may have diminished over time. This would also corroborate the single result.

Author response: The reviewer raises a very interesting point. Following his/her recommendation, we have explored the heme binding capability of a substantial number of additional glycosidases, including four modern bacterial glycosidases, along with five proteins corresponding to sequential nodes in the evolutionary line that leads

from our extensively characterized ancestral glycosidase to the modern glycosidase from *Halothermothrix orenii*. We assessed the amount of heme that is bound to these proteins upon following the standard purification protocol with the metabolic precursor of heme added to the culture medium. In the revised manuscript, these results are given in figures S20 and S21, and discussed on pages 12-13. The capability to recruit heme from overexpression in *E. coli* is highest for our extensively characterized ancestral glycosidase, but decreases substantially in the line of descent that leads to the modern glycosidases. Still, the modern enzymes retain some capability to recruit heme, as is the case with the intermediate ancestral nodes studied. It is clear that our extensively characterized glycosidase ancestor is not an outlier, although it appears that the capability to bind heme became substantially degraded early in the evolution of family 1 glycosidases.

The authors speculate that extant glycosidases may have vestigial heme-binding characteristics, but did they assay any for binding? Alternatively, the authors could generate simple error-prone libraries of the ancestral enzyme to determine how easily the role of heme could be replaced by side chain dynamics. In a reciprocal experiment, extant glycosidases could be randomized to determine whether they could bind heme.

Author response: The preparation and analysis of error-prone libraries, as suggested by the reviewer as an alternative to the assay of vestigial heme-binding in modern glycosidases, is certainly very interesting, but also likely to be extremely time consuming. However, following the first reviewer suggestion, we have included in the revised version (Figure S20) experimental studies of four different modern family 1 glycosidases that clearly reveal their vestigial heme-binding characteristics.

The authors did not elaborate on how the heme-binding motif in the ancestor differs in extant enzymes.

Author response: We thank the reviewer for pointing out this oversight. In the revised version (page 10) we now discuss how the features of the ancestral heme binding pocket compare with extant heme proteins. We note that, as is the case with modern proteins, the ancestral heme binding pocket is enriched in hydrophobic and aromatic residues and propionate anchoring is achieved through interactions with arginine, tyrosine and lysine residues. We also note that tyrosine is not the most common axial ligand but that, nevertheless, there are examples of modern heme proteins that use tyrosine as the axial ligand (e.g. catalases).

I would have appreciated a more in depth discussion on heme's role in catalysis.

Author response: We actually do not think that the heme has a direct role in catalysis (as described on pages 10, 11 and 14) because, among other factors, the heme does not have access to the active site where glycosidase hydrolysis takes place according to the 3D-structure (see Figure S17 in the revised version). Our interpretation (see pages 10, 11 and 14 in the revised version) is that the catalysis enhancement brought about by heme is an allosteric effect linked to changes in the conformational dynamics. This seems to be further supported by the results of the molecular dynamics simulations

shown in Figure 3, which illustrate that the ancestral protein has greater flexibility without heme bound than with heme bound (both overall and in key regions, as discussed in the main text), although even the ancestral glycosidase with heme bound is not as rigid as the corresponding modern enzyme.

I have listed further points below:

- S3 *Thermotoga maritima* misspelled

Author response: Thank you for catching this typo. It has been corrected in the revised version.

- F3 needs key for color coding

Author response: A new version of figure 3 has been provided, with an updated caption.

- F4 and Table S3 and all assay figures – please list replicates (i.e. biological and technical – I could not find this information in the methods either)

Author response: Prompted by one of the comments by referee 3, we have repeated the catalytic experiments for the ancestral and modern glycosidases using extended ranges of pNP-glu and pNP-gal substrates. The results are very similar to and convey the same messages as the values we reported in the first version. For each enzyme-substrate combination, we performed triplicate experiments involving proteins from at least two different preparations. The Michaelis plots for all the replicate experiments are given in the revised version (Figures 4 and S6-S9), the Michaelis-Menten parameters derived from the individual replicas are collected in Table S3 and the average values are given in Figure 4 and in Table S4.

- F4 – include another graph of k_{cat}/K_M s. It is useful to see how turnover rate and Michaelis constants differ between the enzymes, but selective pressure acts at the level of the organism and therefore metabolic flux --- of which k_{cat}/K_M is a large component -- is more important. Showing just these two graphs ablates the fact that the ancestral glycosidase has essentially the same k_{cat}/K_M for pNP-glu in the presence and absence of heme. The increase in k_{cat} and K_M cancel each other out.

Author response: The graph for k_{cat}/K_M is included in Figure 4 of the revised version. The reviewer is correct that there is cancelation between k_{cat} and K_M for the pNP-glu substrate, but this does affect the fact that the hydrolysis rate is higher with the heme-bound protein in most of the studied substrate concentration ranges.

- The text mentions the proficiency of glycosidases. Please include uncatalysed rate and proficiency for each activity. This is particularly useful when discussing the poor activities of ancestral enzymes.

Author response: We have followed the reviewer's suggestion and have included in the revised manuscript estimates of the uncatalyzed rate and the rate enhancement achieved by the ancestral enzyme (page 7). However, some comments regarding these estimates are pertinent here. Glycosidic bonds are extremely stable, even more stable than other covalent bonds in biological polymers, as Richard Wolfenden noted many years ago (ref. 15 in the manuscript). They are so stable that their uncatalyzed hydrolysis can only be easily observed at high temperature. Estimating their uncatalyzed reaction rate at room temperature is, therefore, highly challenging. Wolfenden determined their rate of hydrolysis at high temperatures (above 100°C in sealed quartz tubes with overpressure to avoid boiling) and performed an Arrhenius extrapolation to room temperature. Actually, this is the same approach that he used for other chemical reactions of biological relevance and that provided the basis for his highly regarded work on the extreme difficulty of many biochemical reactions in the absence of enzymes [for instance: Wolfenden, R. Degrees of difficulty of water-consuming reactions in the absence of enzymes. *Chem. Rev.* 106, 3379-3396 (2006)]. Wolfenden found (from Arrhenius extrapolation) a first-order rate constant for the uncatalyzed hydrolysis of β -methylglucopyranoside of $4.7 \cdot 10^{-15} \text{ s}^{-1}$ at 25 °C, which corresponds to half-life at 25 °C on the order of millions of years. Since the k_{cat} values for our ancestral glycosidase are about 10^{-1} s^{-1} , we have a rate acceleration of about 13 orders of magnitude. Of course, this is a rough estimate because the uncatalyzed rate comes from an Arrhenius extrapolation and corresponds to a β -glucopyranoside which is somewhat different than the one we are using (methyl vs. PNP as the aglycone moiety). The important point is that our ancestral glycosidase is *not* a poor enzyme. It is simply not as good as many modern glycosidases, but it still accelerates the rate of glycoside bond hydrolysis by many orders of magnitude (whether it is 13 orders of magnitude or 12 or 14 does not change this conclusion). This is a point that we failed to clearly make in the first version of the manuscript but that we emphasize in the revised version (page 7). We thank the reviewer for bringing up this important issue.

- S5 comment on poor saturation of enzymes with pNP-gal – were higher concentrations assayed or were KM values extrapolated from the data? What is the limit for pNP-gal solubility in the assay?

Author response: In view of the reviewer comment, we have repeated all kinetic determinations with pNP-gal and pNP-glu using substantially extended substrate concentration ranges, up to 20 mM for the pNP-glu substrate and up to 50 mM for the pNP-gal substrate (see Figures 4 and S6-S9 in the revised version). Using wide substrate concentration ranges poses several challenges. First, the experimental protocol needs to be fine-tuned to ensure that the dilution steps involved do not bring about changes in solvent composition that could distort the profiles. Second, as it has been described in the literature (see, for instance, Kuusk, S. & Väljamäe, P. *Biotechnol. Biofuels* 10:7, 2017), glycosidase catalysis often shows kinetic complexities at high substrate concentrations, due to phenomena such as transglycosylation or inhibition by substrate. As a result of these complexities, Michaelis-Menten saturation kinetics are sometimes not observed with wider substrate concentration ranges. In our experiments, we found this non-saturation pattern only with the pNP-glu substrate and some of the modern proteins we tested (Figures S6-S9). In these cases, only the

data up to a substrate concentration of 5-8 mM were used for the determination of the catalytic parameters from the fits of the Michaelis-Menten equation. It must be noted that for each enzyme-substrate combination, we performed triplicate experiments involving proteins from at least two different preparations. We hope that the effort we have made to provide reliable catalytic parameters is appreciated. We note, nevertheless, that the catalytic parameters given in the revised version (Figure 4) are very similar and convey the same messages as the values we reported in the first version.

- The paper describes the ancestral enzyme as a generalist because it demonstrates mM-range K_M s for both pNP-glu and pNPgal. However, the ancestor has a k_{cat}/K_M for pNP-glu that is 6-fold higher than that for pNP-gal. *H. orenii* has a 7-fold higher k_{cat}/K_M for pNP-glu vs pNP-gal. Is the ancestor really a generalist? Why aren't the extant enzymes more specialized?

Author response: We never intended to make a grand statement regarding the generalist nature of the glycosidase ancestor. We are sorry if perhaps our reference to Jensen paper (reference 27 in the first version) conveyed the wrong message and we thank the reviewer for raising this issue. All we wanted to point out is that there is well-known pattern in the Michaelis constants for modern family 1 glycosidases (quoting from the GH1 article in CAZypedia written by Stephen Withers: "The most common known enzymatic activities for glycoside hydrolases in this family are β -glucosidases and β -galactosidases: indeed typically both activities are found within the same active site, often with similar k_{cat} values, but with substantially higher K_M values for the galactosides") and that this pattern of Michaelis constant values is not observed in our ancestral glycosidase, which seems consistent with an early stage in the evolution of family 1 glycosidases. We pointed this out in the context of differentiating between the ancestral properties that appear reasonable for an early evolutionary stage versus those (heme binding) that are truly shocking.

We think that the substrate scope of the ancestral glycosidase is much clearly expounded upon in the revised version because we have added experimental catalysis data for a large number of different substrates (see below). Also, we have eliminated the reference to the old Jensen paper which, although a fundamental reference in the field of enzyme evolution, it may be misleading in the specific context of the work.

- If specialization occurs at the level of K_M , then one could argue that the ancestor does not have physiologically-relevant activity for either substrate especially in the presence of heme, which increases the K_M for both substrates.

Author response: It is not clear to us that the K_M values can be used to argue against the physiological relevance of the activities of the ancestral glycosidase. Certainly, the relation between the Michaelis constant and the physiological substrate concentration determines the kinetic behaviour of the enzyme *in vivo* (whether the rate is responsive to substrate concentration or buffers changes in substrate concentration). However, we do not know the ancestral physiological concentrations for the substrates, and we do not see how the physiological relevance of the activities can be discussed on the

basis of the K_M values alone. For instance, heme binding could bring the K_M values close to the physiological substrate concentrations thus enabling regulation. However, this, while plausible, is speculative because we do not know the ancestral substrate concentrations.

- Did the authors assay the ancestor with more disparate substrates?

Author response: Yes, we did. These extensive data are added and described in the revised version. For the convenience of the reviewer, we reproduce the description here:

1) Using the same methodology employed with 4-nitrophenyl- β -D-glucopyranoside and 4-nitrophenyl- β -D-galactopyranoside (Figure 1), we determined profiles of rate versus temperature for the ancestral glycosidase and the modern glycosidases from *Halothermotrix orenii* and *Sacharophagus degradans* using as substrates 4-nitrophenyl- β -D-fucopyranoside, 4-nitrophenyl- β -D-lactopyranoside, 4-nitrophenyl- β -D-xylopyranoside and 4-nitrophenyl- β -mannopyranoside. In all cases (Figure S11 of the revised version) we found the levels of catalysis of the ancestral protein to be depressed in comparison with the modern proteins. We also found that the levels of catalysis for the β -D-glucopyranoside and β -D-fucopyranoside substrates were similar, but this pattern is also observed with the modern proteins.

2) We carried single activity determinations at 25 °C for the ancestral glycosidase with a wider range of substrates, including derivatives of disaccharides (maltose, cellobiose) and several substrates with an α anomeric carbon (Table S6 of the revised version); however, we did not find any substrate with a catalysis level substantially higher than those previously determined and, in many cases (in particular with the α substrates), no significant activity was detected.

3) Since some of the proteins under the N72 node are 6-phosphate- β -glucosidases (Figures 1A and S1), we tested the activity of our ancestral glycosidase against 4-nitrophenyl- β -D-glucopyranoside-6-phosphate (Figure S12); however, we found a catalytic efficiency ~40 fold smaller than that determined with the corresponding non-phosphorylated substrate.

4) Glycosidases are typically described as being very promiscuous for the aglycone moiety of the substrate (the part of the substrate that is replaced with *p*-nitrophenyl in the substrates commonly used to assay glycosidase activity) while they are more specialized for the glycone moiety of the substrate. However, the flexibility in certain regions of the ancestral structure could perhaps favor the hydrolysis of substrates with larger aglycone moieties. To explore this hypothesis, we tested four synthetic substrates with aglycone moieties larger than the usual *p*-nitrophenyl group (Figure S13). Still, we found levels of catalysis substantially depressed with respect to those obtained for the modern glycosidase from *Halothermothrix orenii*, used here as comparison.

Overall, it appears reasonable that our resurrected ancestral enzyme reflects an early stage in the evolution of family 1 glycosidases, at which catalysis was not yet optimized

and substrate specialization had not yet evolved. Of course, the number of different glycosidase substrates is overwhelming, as an examination of the CAZypedia resource immediately shows. Therefore, we cannot absolutely rule out that the ancestral glycosidase is highly efficient for some substrate we have not tested. However, testing all glycosidase substrates is hopefully out of the question for our study.

It is also important to note that, for most additional substrates included in the revised version, we have not determined full Michaelis profiles. There are several reasons for this. Firstly, we mainly wanted to explore the possibility that some substance could be a much better substrate of the ancestral glycosidase than the common β -D-glucopyranoside and β -D-galactopyranoside substrates. Secondly, determining Michaelis plots for all these substrates would have required an overwhelming amount of work and money, since some of the substrates we have tested are prohibitively expensive.

- Are the residues that interact with heme conserved in the extant proteins?

Author response: They are conserved to some significant extent and, in fact, the ancestral residues are the consensus residues in the set of modern sequences used as a starting point for reconstruction. Still, conservation is far from strict and the sequences of modern glycosidases in the set differ from the ancestral sequence at many of the positions involved in heme interactions in the ancestral protein. These statistical analyses are provided in the Tables S7 and S8 and briefly mentioned in the main text of the revised manuscript (page 10).

- Fig 6 has too many panels. Include A, B and a version of D or E.

Author response: In the revised version, Figure 6 only includes panels A, B and C, while panels D and E of the previous version of the figure have been moved to Supplementary Information (Figure S17).

- How many types of glycosidases are predicted to have been present in LUCA?

Author response: 355 protein families are inferred to have been in LUCA: Weiss, C.W., Sousa, F.L., Mrnjavac, N., Neukirchen, S., Roettger, M., Nelson-Sathi, S. & Martin, W.F. The physiology and habitat of the last universal common ancestor. *Nature Microbiol.* **1**, 16116 (2016).

Out of these, 19 are sugar-related and include glycosidases GH15 and GH1 (that is, the family 1 glycosidases we have studied in this work using ancestral reconstruction). This information is briefly provided in the revised version (page 4).

Reviewer #3 (Remarks to the Author):

This manuscript describes a very intriguing result from ancestral sequence reconstruction whereby the process of reconstruction has “discovered” a heme binding activity for an ancient GH1 family glycosidase enzyme. The authors describe a

range of biochemical experiments to demonstrate the stoichiometric binding of heme and the relationship between binding and enzymatic activity. Molecular dynamics simulations are additionally described to model the dynamics of the ancestral structures (as determined by crystallography).

The conundrum for the reader is whether this is an interesting oddity or whether heme binding is a significant evolutionary intermediate on the trajectory (over deep time) for GH1 TIM barrel glycosidase enzymes. It appears that heme binding is robust and stoichiometric. The fact that heme binding improves k_{cat} (although not k_{cat}/K_m) is further strong evidence for this observation being significant. However, some of the assertions made by the authors are rather weak and deserve further analysis. In my opinion, major revision may provide further evidence that strengthens the case for this being a significant evolutionary finding.

Author response: The issue raised by the reviewer (whether heme binding is an interesting oddity or a significant evolutionary intermediate) is certainly crucial. Following his/her recommendation, we have additionally explored the heme binding capability of a substantial number of additional glycosidases, including four modern bacterial glycosidases and five proteins corresponding to the sequential nodes in the evolutionary line that leads from our extensively characterized ancestral glycosidase to the modern glycosidase from *Halothermothrix orenii*. We assessed the amount of heme that is bound to these proteins upon following the standard purification protocol with the metabolic precursor of heme added to the culture medium. In the revised version, these results are given in Figures S20 and S21, and discussed on pages 12-13. The capability to recruit heme from overexpression in *E. coli* expression is highest for our extensively characterized ancestral glycosidase, and decreases substantially in the line of descent that leads to the modern glycosidases. Therefore, the heme is not binding to these scaffolds indiscriminately. However, the modern enzymes retain some capability to recruit heme, as is the case with the intermediate ancestral nodes studied. It is clear that our extensively characterized glycosidase ancestor is not an outlier, although it appears that the capability to bind heme becomes substantially degraded early in the evolution of family 1 glycosidases.

Major points:

The structure for the apo-ancestral enzyme has two loops missing due to disorder. These loops become (mostly) ordered upon heme binding. The authors rightly state that no extant GH1 enzymes display heme binding. Thus, the observation of a heme binding site is surprising and very interesting. They further speculate that a heme binding module may have been grafted onto the TIM barrel during evolution. This raises several questions:

1. The statement in Line 395 “flexibility is greatly enhanced and extends to large parts of the structure” is reiterated in several places in the manuscript. However, is it not more accurate to say that there are two flexible loops in the structure and that the effect is localised to these loops?

Author response: We agree that it is more accurate to state the specific loops and we do so in several parts of the revised manuscript.

2. From a cursory look at the PDB, the extant thermophilic homologue from *Halothermothrix orenii* that is used as a comparison throughout the manuscript has a large PEG molecule bound to one of these “flexible” loops. Other closely related structures have e.g. hydrophobic groups bound in this region (see 2WC4 with a substrate analogue). Thus, it may be the case that the loops in question are “binding loops” in a general sense. A thorough review of the GH1 family enzymes will reveal this.

Author response: This is a very interesting possibility and we thank the reviewer for pointing this out. Following his/her suggestion, we have reviewed the 3D structures of bacterial GH1 enzymes reported in CAZy (excluding some obvious redundancies) and found that 75 ions or small molecules bound to the regions corresponding to the flexible regions in our ancestral glycosidase. However, we also find 163 ions and small molecules bound to other regions of the protein structure. It is not clear to us that these numbers provide a clear statistical support for the hypothesis that the loops in question are binding loops in the general sense. Therefore, we have not included a discussion on this hypothesis in the revised version of the manuscript.

Further, many GH enzymes have auxiliary domains. Are these loops common locations for the insertion of auxiliary domains?

Author response: As far as we know, these auxiliary domains are often carbohydrate binding domains and, more specifically, cellulose binding domains in many cases. However, cellulose degradation is not among the activities typically described for family 1 glycosidases. Of course, there may be an evolutionary relationship between the flexible loops we observe in the ancestral GH1 and the location of auxiliary domains in other glycosidase families that display cellulase activity (GH5, for instance). However, we believe that this possibility, although interesting, is highly speculative, given the difficulties of establishing common ancestry between different glycosidase families.

3. Related to 2. the authors state on line 437: “Albeit limited, our results suggest that some modern family 1 glycosidases may retain a vestigial capability to bind heme.” This is a very intriguing claim that deserves further analysis (see 4. below).

Author response: As mentioned in our response to a previous suggestion above, we have included in the revised version (Figure S20) experimental studies of four different modern family 1 glycosidases that clearly reveal their vestigial heme-binding characteristics. The results are discussed on pages 12-13 of the revised manuscript.

4. If the heme binding module has been grafted onto the TIM barrel, then it should be possible to find evidence for this hypothesis. For example, the authors state that an early fusion event with a heme-containing domain is a possibility. This can be

tested. Is there any evidence for a module from the heme binding proteins in the PDB? There are numerous approaches to identify modular components of structures related to binding to test this assertion.

Author response: This is a very interesting issue and we thank the reviewer for bringing it up. To explore the possibility suggested by the reviewer, we used the DALI server to search the Protein Data Bank for structural alignments of the alpha-helices involved in heme binding in our ancestral glycosidase. 222 alignments were returned with RMSD ranging from 1.6 to 11.4 Å. However, only 3 of the proteins had heme bound and the structural similarity with the query structure was actually poor (see page 13 and Figure S22 in the revised manuscript). It is possible that the structure of the ancestral heme-containing domain was distorted upon fusion and subsequent evolution, and, therefore, it is difficult to identify in searches of modern protein structures. Another possibility that we note in the revised version of the manuscript (page 13) is that there was never a fusion event and heme was already present even at the most ancient stages in the early evolution of family 1 glycosidases. This hypothesis would be consistent with the notion that cofactors are molecular fossils and that they may have facilitated the primitive emergence of proteins by selecting them from a random pool of polypeptides (see ref. 50 in the revised version).

5. The molecular dynamics experiments deserve a greater profile in the manuscript in my opinion. They appear to indicate some very interesting trends (Figure S9). For example, the starting structures for the ancestral sequences appear to be a long way from the equilibrium structure. What is the relationship between the starting structure (presumably the crystal structure) and the equilibrium structures for the dynamics trajectories? It is interesting that the contemporary crystal structure for *Halothermothrix orenii* is much closer to the equilibrium MD ensemble. The standard deviation for the RMSD for the heme-bound structure is greater than that for the non-heme bound structure. What is the reason for this (somewhat counterintuitive) result? Does this shed light on the relationship between heme binding and catalysis?

Author response: We have now included the following clarifications on page 10 of the manuscript:

Heme binding clearly rigidifies the ancestral protein, as shown by fewer missing regions in the electronic density map, in contrast to the structure of the heme-free protein (see Figures 2A, 2B and 2C). This is also confirmed by molecular dynamics simulations of the ancestral glycosidase both with and without heme bound (Figures 3 and S18). Figure S18 shows the backbone RMSD (Å) over ten individual 500 ns MD simulations per system, and, from this data, it can be seen that while the RMSD is fairly stable in the case of the modern protein, the ancestral glycosidases (both with and without heme bound) are initially quite far from their equilibrium structures, due to the high flexibility of the missing regions of the protein which require substantial equilibration. In addition, we note that while the overall average RMSD for the ancestral protein with heme bound is slightly lower than for the ancestral protein without heme (Figure S18), the standard deviation is higher. This is due to the greater flexibility of the reconstructed missing loop (see the Methods section), which allows it

to sample a larger span of conformations depending on whether the loop is interacting with the bound heme or not (we observe both scenarios in our simulations of the heme-bound ancestral glycosidase). In contrast, in the absence of the heme, the loop is always in a flexible open conformation leading to a higher overall RMSD but a lower standard deviation as a narrower range of conformations are sampled in our simulations. As neither the loop nor the heme have access to the active site (Figure S17), these differences are unlikely to have a direct effect on catalysis.

6. Figure 2D is not terribly informative without reference to table S4. Is it possible to label the residues in figure 2D?

Author response: Yes, of course. The residues in Figure 2D are now labelled in the revised version.

The enzymatic assays are well done, however the analyses need some revision in my opinion. The authors cite Wolfenden and state that glycosidases accelerate the chemical reaction by up to 17 orders of magnitude. In light of this, a value for k_{cat} of $\sim 0.1 \text{ s}^{-1}$ still represents a rate enhancement of ~ 15 orders of magnitude.

Author response: This is a good point. In the revised version (page 7), we provide Wolfenden's estimate of the rate of the uncatalyzed reaction and emphasize that the rate enhancement of many orders of magnitude achieved by the ancestral glycosidase and, therefore, that it cannot be considered as a "poor enzyme".

Hence, considering the statement "the comparatively low activity of the ancestral protein is likely linked to its conformational flexibility" raises some questions given that the differences in rate are very subtle when compared to the rate enhancement which remains extremely high in both cases (i.e. for both gluco and galacto substrates). There is no evidence that the conformational flexibility is influencing the very subtle differences in rate. Can the authors point to some evidence for this?

Author response: The main evidence for a role of conformational flexibility in the rate variations comes from the fact that the 3D-structures do not show significant differences at the active site positions, as we note in the manuscript (page 11 and Figure 2E). Hence, the observed rate differences are likely related to factors that are not apparent in static X-ray structures, i.e., to factors related to dynamics. In addition, we have now examined the RMSF values of key catalytic residues and found that the flexibility of several of these residues is reduced upon moving from the ancestral glycosidases without heme, to adding the heme, to the modern glycosidase, in a clear decreasing trend (see page 11 and Figure S19 in the revised manuscript).

The kinetics for 4-PNP-glucopyranoside are intriguing. k_{cat} differs by a factor of 3.4 whereas k_{cat}/K_m values are approximately equal when apo- and heme-bound enzymes are compared. Thus, heme binding is significantly impairing K_m . Indeed, a K_m value of $\sim 9 \text{ mM}$ warrants some discussion. This is approaching a value commensurate with non-specific binding. This suggests to me that glucose (and galactose) are not the preferred substrates for this ancestral enzyme.

Author response: As mentioned earlier, it is not clear to us that the K_M values can be used to argue against the physiological relevance of the activities of the ancestral glycosidase. Certainly, the relation between the Michaelis constant and the physiological substrate concentration determines the kinetic behaviour of the enzyme *in vivo* (whether the rate is responsive to substrate concentration or buffers changes in substrate concentration). However, we do not know the ancestral physiological concentrations for the substrates, and we do not see how the physiological relevance of the activities can be discussed on the basis of the K_M values alone. For instance, heme binding could bring the K_M values close to the physiological substrate concentrations thus enabling regulation. However, this, while plausible, is speculative because we do not know the ancestral substrate concentrations. Overall, we don't think that we can rule out the common β -glucopyranosides and β -galactopyranosides as the preferred substrates for the ancestral enzyme, inasmuch as our experimental studies with a substantial number of additional substrates (see below) do not suggest any better candidates.

Have the authors searched for alternative substrates?

Author response: Yes, we have. These extensive additional data are now included and described in the revised version. For the convenience of the reviewer, we reproduce the description here:

1) Using the same methodology employed with 4-nitrophenyl- β -D-glucopyranoside and 4-nitrophenyl- β -D-gaactopyranoside (Figure 1), we determined profiles of rate versus temperature for the ancestral glycosidase and the modern glycosidases from *Halothermotriz orenii* and *Sacharophagus degradans* using as substrates 4-nitrophenyl- β -D-fucopyranoside, 4-nitrophenyl- β -D-lactopyranoside, 4-nitrophenyl- β -D-xylopyranoside and 4-nitrophenyl- β -mannopyranoside. In all cases (Figure S11 of the revised version) we found the levels of catalysis of the ancestral protein to be depressed in comparison with the modern proteins. We also found that the levels of catalysis for the β -D-glucopyranoside and β -D-fucopyranoside substrates were similar, but this pattern is also observed with the modern proteins.

2) We carried single activity determinations at 25 °C for the ancestral glycosidase with a wider range of substrates, including derivatives of disaccharides (maltose, cellobiose) and several substrates with an α anomeric carbon (Table S6 of the revised version); however, we did not find any substrate with a catalysis level substantially higher than those previously determined and, in many cases (in particular with the α substrates), no significant activity was detected.

3) Since some of the proteins under the N72 node are 6-phosphate- β -glucosidases (Figures 1A and S1), we tested the activity of our ancestral glycosidase against 4-nitrophenyl- β -D-glucopyranoside-6-phosphate (Figure S12 of the revised version); however, a found a catalytic efficiency ~40 fold smaller than that determined with the corresponding non-phosphorylated substrate.

4) Glycosidases are typically described as being very promiscuous for the aglycone moiety of the substrate (the part of the substrate that it is replaced with *p*-nitrophenyl in the substrates commonly used to assay glycosidase activity) while they are more specialized for the glycone moiety of the substrate. However, the flexibility in certain regions of the ancestral structure could perhaps favor the hydrolysis of substrates with larger aglycone moieties. To explore this hypothesis, we tested four synthetic substrates with aglycone moieties larger than the usual *p*-nitrophenyl group (Figure S13). Still, we found levels of catalysis substantially depressed with respect to those obtained for the modern glycosidase from *Halothermothrix orenii*, used here as comparison.

Overall, it appears reasonable that our resurrected ancestral enzyme reflects an early stage in the evolution of family 1 glycosidases, at which catalysis was not yet optimized and substrate specialization had not yet evolved. Of course, the number of different glycosidase substrates is overwhelming, as an examination of the CAZypedia resource immediately shows. Therefore, we cannot absolutely rule out that the ancestral glycosidase is highly efficient for some substrate we have not tested. However, testing all glycosidase substrates is hopefully out of the question for our study.

It is also important to note that, for most additional substrates included in the revised version, we have not determined full Michaelis profiles. There are several reasons for this. First, we mainly wanted to explore the possibility that some substance could be a much better substrate of the ancestral glycosidase than the common β -D-glucopyranoside and β -D-galactopyranoside substrates. Second, determining Michaelis plots for all these substrates would have required an overwhelming amount of work and money, since some of the substrates we have tested are prohibitively expensive.

It is certain that the natural substrates are not 4-PNP-gluco(galacto)pyranosides.

Author response: The reviewer is correct, of course, as the PNP moiety is not natural and it is included just for easy detection of the hydrolysis. However, as noted above, glycosidases are typically described as being very promiscuous for the aglycone moiety of the substrate while they are more specialized for the glycone moiety of the substrate (see ref. 14 in the revised version). Therefore, PNP is used as the aglycone in the typical substrates used to test glycosidase activity because the chemical structure at the aglycone portion is not expected to be crucial for catalysis. Still, as we have mentioned above, we also considered the possibility that the enhanced flexibility of the ancestral protein allowed for a more efficient catalysis of substrates with bulky aglycones as compared with modern glycosidases (Figure S13).

Figure 5C needs some explanation. What is the curve fitted to the 1 microM data? Why has the curve not been simultaneously fitted to the 100 nM and 50 nM data?

Author response: A realistic model for the heme binding kinetics is not straightforward to derive, among other things because heme in solution at neutral pH associates with time thus decreasing the amount of monomeric heme (the form more competent for binding). Therefore, the curve shown was an exponential-based fit to the data meant

only to guide the eye. In the revised version, we have included such fits also for the 100 nM and 50 nM data.

The caption states that heme “binding is strong”. What is the binding constant for heme?

Author response: Since heme affects activity even at very low concentrations, the binding is assured to be strong. However, in the revised version we have included a direct determination of the dissociation constant using Microscale Thermophoresis, a technique commonly used to characterize interactions between biomolecules. A value of the heme dissociation constant of 547 ± 110 nM from three independent determinations (Figure S14 in the revised version). Actually, as we note in the revised version (see caption of Figure S14), this value is likely to be an overestimate (i.e., the actual dissociation constant could be even smaller) because of the already mentioned problem of heme association at neutral pH. Overall, there is no doubt that the binding is strong, with a dissociation constant in the submicromolar range.

Citing a paper from 1976 to justify the statement that ancient enzymes are generalists is not convincing.

Author response: This is good point. In the revised version, the issue of the ancestral promiscuity is discussed in more realistic terms (page 7-8) and also cite more recent publications on the subject (refs. 5, 7, 9 and 29 in the revised manuscript).

In summary, this manuscript describes a very interesting and potentially significant finding - heme binding in a reconstructed GH1 enzyme. However, in my opinion, there are several hypotheses and speculative statements presented with relatively weak evidence or without evidence. This can potentially be rectified by further analyses.

Minor points:

Figure 2. The homology model in Figure 2A can be removed as it does not add anything to the analyses.

Author response: The reason for providing the homology model is related to the missing regions in the experimental structure, which correspond to high flexibility loops. This important result is made visually apparent upon a comparison with the homology model which, of course, includes the whole protein. We are sorry that we this was not clear in the first version of the manuscript. In the revised version, the point is clearly made in the main text (page 6 and caption to Figure 2).

Line 383 “The TIM-barrel is the most common protein fold” - I don’t think that this is the case. Perhaps the most common enzyme fold?

Author response: This has been corrected.

Table S6 - Rmerge for ancestral-heme is incorrect.

Author response: Thanks for catching the typo. It has been corrected in the revised version.

Figure 3 caption... “[Adrian describe clustering algorithm...]”?

Author response: We apologize for this oversight. The text is a leftover from the interaction between two of the authors...The comment has been eliminated from the revised version and the description of the clustering algorithm has been included.

Reviewer #4 (Remarks to the Author):

This manuscript provides an interesting account of an ancestral sequence reconstruction (ASR) experiment that has yielded a glycoside hydrolase (GH) with the curious ability to bind heme. This appears to be the first report of a heme-binding GH. Heme binding increased this GH’s activity (kcat/Km) by a factor of three. It is proposed that this heme-binding ASR-generated GH may approximate an early stage in the evolution of the GH1 family and that this sequence may represent a useful chassis for the directed evolution of GHs with novel activities or allosteric activation mechanisms. However, these ideas are not explored in this preliminary investigation.

While the heme-binding of this GH is intriguing, I have two main reservations about the significance of this work:

1) It remains unclear if heme-binding is a characteristic shared by many ASR-generated sequences from different nodes of the GH1 phylogenetic tree, or if this is merely an artefact observed for this particular sequence. Having established a structural basis for heme binding, the authors could easily perform sequence alignments with ASR sequences from other nodes and even modern sequences to determine if the heme-binding motif is conserved. If so, it would be good to express and characterise these putative heme-binding proteins too. That would prove that this phenomenon is unlikely to be an artefact. The authors did initially explore three ASR sequences (nodes 72, 73 and 125) before abandoning N73 and N125 due to their propensity to aggregate...it is unclear if they ever tried rescuing these proteins by the addition of heme. More work is needed to establish the significance of heme-binding by this single ASR protein.

Author response: We fully agree with the reviewer and, in fact, other reviewers have expressed similar points of view. Therefore, to clarify this issue we have now explored the heme binding capability of a substantial number of additional glycosidases, including four modern bacterial glycosidases and five proteins corresponding to the sequential nodes in the evolutionary line that leads from our extensively characterized ancestral glycosidase to the modern glycosidase from *Halothermothrix orenii*, as well as the N73 and N125 nodes. We assessed the amount of heme (if any) that is bound to these proteins upon following the standard purification protocol without and with the metabolic precursor of heme added to the culture medium. In the revised version, these additional results are presented in Figures S20 and S21, and discussed on pages

12-13. The capability to recruit heme from the *E. coli* expression is highest for our extensively characterized ancestral glycosidase, but decreases substantially in the line of descent that leads to the modern glycosidases. Still, the modern enzymes retain some capability to recruit heme when the culture medium is supplemented with the metabolic precursor of heme. Preparation of the heme-saturated forms of the modern glycosidases and the intermediate nodes is challenging, given their limited affinity for heme. Still, it is clear that our extensively characterized heme binding to the glycosidase ancestor is not an artefact observed for a particular sequence, as heme binding is observed for other ancestral nodes and, in a clearly vestigial form, in several modern family 1 glycosidases.

2) The activation upon heme binding is very modest – a factor of three. While this subtle effect is claimed to be useful as a starting point for biosensor development, this idea remains an unrealised aspiration.

Author response: The reviewer comment makes us realize that, in the original version, we fail to make clear the meaning and implications of the effect of heme binding on catalysis. Directed laboratory evolution, the methodology that was awarded the 2018 Chemistry Nobel prize, can be used to enhance or modify any functionality, provided that a certain level of such functionality is available to start the process. That is, evolution (either natural or directed in the laboratory) needs at least a low level of a functionality to act upon and the availability of such “seed” levels may become a critical bottleneck. The *de novo* generation of new functionalities is, therefore, a very important, unsolved problem in protein engineering. Our work uncovers a heme binding capability and a possibility of allosteric regulation that were previously unknown in glycosidase enzymes. That is, our work leads to a novel, relevant functionality at a seed level that can be used as starting point for subsequent directed evolution. In this context, the factor of three is actually quite significant. We believe that all these points are clearly expounded in the revised version (page 13).

Furthermore, it remains unclear how a glucosidase activity might be used as a readout for heme concentrations.

Author response: As shown in panel D of figure 6 in the revised version, the activity of the ancestral glycosidase depends on heme saturation and heme binding is tight. This would provide a way to assess the amount of heme present. Still, the general make point we make in the manuscript (pages 13-14) is that directed evolution plus computational design could potentially be used to engineer binding and a detection capability for other molecules of interest.

A less significant issue I had with the manuscript was that I found the narrative to be somewhat disjointed. The heme-binding nature of this protein appeared very early on with the structural work without being addressed until much later. The authors could perhaps rethink how some of these figures and results are assembled to provide a smoother transition between topics.

Author response: Heme-binding appears early in the manuscript because we knew that the protein binds heme before we did the structural work. The reason is that heme binding is visually obvious, as the ancestral protein becomes reddish. That is, the narrative follows the way the work was actually carried out. We realize, of course, that the crucial piece of information that the protein becomes reddish upon heme binding was perhaps not clearly conveyed in the first version of the manuscript. We have included an additional figure with pictures of protein preparations (Figure 5 in the revised version) which we believe will contribute to make the narrative of the work intuitive and appealing to the reader.

In addition to the above, a few suggestions to improve this manuscript include:
Figure S3. SEC with external calibration is not a reliable way to determine protein oligomerisation state. Please perform SEC-MALS, AUC or SAXS to determine oligomerisation state in solution.

Author response: In the revised version, we added AUC (analytical ultracentrifugation) data that confirm the monomeric nature of our ancestral glycosidase (Figure S4).

Figure 2A. Why include a homology model when you have the actual structure? This doesn't add any value to the work.

Author response: The main reason is that there are missing regions in the experimental structure, which correspond to high flexibility loops. This important result is made visually apparent upon a comparison with the homology which, of course, includes the whole protein. We are sorry that this was not clear in the first version of the manuscript. In the revised version, the point is clearly made in the main text (page 6 and caption to Figure 2).

Figure 2C. Proteolysis is a crude method of assessing a protein's structural dynamics, since increased proteolysis is both fold and sequence related. Please consider using NMR or HDX experiments to more reliably address the question of protein dynamics.

Author response: NMR and isotopic exchange for a protein of the large size of our ancestral glycosidase is far from trivial, is not guaranteed to work due to the large system size and, in any case, it would be highly demanding in terms of required time and effort. We agree, of course, that proteolysis is a crude method to assess conformational flexibility, but in this case, the results from proteolysis agree with the B-factor profiles, the regions missing in X-ray maps and the Molecular Dynamics simulations. We think that our conclusions are robust.

The authors should expand their assessment of the enzyme's substrate preference beyond simple synthetic PNP-glycosides to include biologically relevant substrates like cellobiose, laminaribiose, gentiobiose and lactose.

Author response: Data on an expanded substrate range are now included and described in the revised version. Among the new substrates studied, there are

derivatives of cellobiose and lactose, as well as substrates with aglycone moiety different than the “usual” PNP. For the convenience of the reviewer, we reproduce here the description provided in the revised manuscript:

1) Using the same methodology employed with 4-nitrophenyl- β -D-glucopyranoside and 4-nitrophenyl- β -D-galactopyranoside (Figure 1), we determined profiles of rate versus temperature for the ancestral glycosidase and the modern glycosidases from *Halothermotrix orenii* and *Sacharophagus degradans* using as substrates 4-nitrophenyl- β -D-fucopyranoside, 4-nitrophenyl- β -D-lactopyranoside, 4-nitrophenyl- β -D-xylopyranoside and 4-nitrophenyl- β -mannopyranoside. In all cases (Figure S11 of the revised version) we found the levels of catalysis of the ancestral protein to be depressed in comparison with the modern proteins. We also found that the levels of catalysis for the β -D-glucopyranoside and β -D-fucopyranoside substrates were similar, but this pattern is also observed with the modern proteins.

2) We carried single activity determinations at 25 °C for the ancestral glycosidase with a wider range of substrates, including derivatives of disaccharides (maltose, cellobiose) and several substrates with an α anomeric carbon (Table S6 of the revised version); however, we did not find any substrate with a catalysis level substantially higher than those previously determined and, in many cases (in particular with the α substrates), no significant activity was detected.

3) Since some of the proteins under the N72 node are 6-phosphate- β -glucosidases (Figures 1A and S1), we tested the activity of our ancestral glycosidase against 4-nitrophenyl- β -D-glucopyranoside-6-phosphate (Figure S12); however, we found a catalytic efficiency ~40 fold smaller than that determined with the corresponding non-phosphorylated substrate.

4) Glycosidases are typically described as being very promiscuous for the aglycone moiety of the substrate (the part of the substrate that is replaced with *p*-nitrophenyl in the substrates commonly used to assay glycosidase activity) while they are more specialized for the glycone moiety of the substrate. However, the flexibility in certain regions of the ancestral structure could perhaps favor the hydrolysis of substrates with larger aglycone moieties. To explore this hypothesis, we tested four synthetic substrates with aglycone moieties larger than the usual *p*-nitrophenyl group (Figure S13). Still, we found levels of catalysis substantially depressed with respect to those obtained for the modern glycosidase from *Halothermothrix orenii*, used here as comparison.

Overall, it appears reasonable that our resurrected ancestral enzyme reflects an early stage in the evolution of family 1 glycosidases, at which catalysis was not yet optimized and substrate specialization had not yet evolved. Of course, the number of different glycosidase substrates is overwhelming, as an examination of the CAZypedia resource immediately shows. Therefore, we cannot absolutely rule out that the ancestral glycosidase is highly efficient for some substrate we have not tested. However, testing all glycosidase substrates is hopefully out of the question for our study.

It is also important to note that, for most additional substrates included in the revised version, we have not determined full Michaelis profiles. There are several reasons for this. First, we mainly wanted to explore the possibility that some substance could be a much better substrate of the ancestral glycosidase than the common β -D-glucopyranoside and β -D-galactopyranoside substrates. Second, determining Michaelis plots for all these substrates would have required an overwhelming amount of work and money, since some of the substrates we have tested are prohibitively expensive.

Since the N72 node also includes enzymes active on Glc-6-P, the authors should also assess this ASR enzyme for activity on this PNP-Glc6P. I appreciate that it is not commercially available but it is easy enough to prepare using hexose kinase.

Author response: Following the reviewer's suggestion, we have prepared PNP-Glc6P, although we used a chemical synthesis procedure, rather than an enzymatic synthesis (actually, collaborators in the Organic Chemistry Department did the synthesis for us: these scientists are now included in the authors list of the revised version). The results, however, indicate a very low activity of the ancestral protein for this substrate (Figure S12 of the revised version). We also studied a PNP-Gal6P substrate (which is commercially available) and obtained the same kind of experimental results (Table S6 of the revised version).

It would be good to see some MS data for heme from the purified protein to support the structural assignment.

Author response: Following the reviewer's advice, we have performed MS experiments with the purified ancestral protein with and without heme bound. The results do support the structural assignment. They are given in Figure S15 of the revised version and briefly mentioned in the main text (page 9).

Also, no attempt appears to have been made to measure the affinity of heme for this protein. This should be rectified.

Author response: It has been rectified. In the revised version, we have included a direct determination of the dissociation constant using Microscale Thermophoresis, a technique commonly used to characterize interactions between biomolecules. A value of the heme dissociation constant of 547 ± 110 nanomolar from three independent determinations (see Figures S14 in the revised version where the validation reports are also included). Actually, as we note in the revised version (see caption to Figure S14), this value is likely to be an overestimate (i.e., the actual dissociation constant could be even smaller) because of the already mentioned problem of heme association at neutral pH. Overall, there is no doubt that the binding is strong, with a dissociation constant at least in the submicromolar range.

6) Figure 6C-E was all a little unclear to me. Some or all of these images need to be enlarged, better annotated and moved to the SI. Figure 6E – in silico mutation work – is highly conjectural and is probably best removed altogether.

Author response: Figure 6 has been modified in the revised version. We only keep panels A, B and C of the previous versions, panel C has been enlarged and the residue annotations in panels A and B have been enlarged. We hope that the Figure is now more informative. We agree that panel E is conjectural, but we believe that many readers will find it useful for illustration. Thus, we have moved it (together with panel D) to a new figure in Supplementary Information (Figure S17)

Minor points

Line 207-208. Is 2.5Å 'good' resolution?

Author response: We did not explain clearly what we meant. We only meant that 2.5 Å is a resolution good enough to be able to trace the course of the polypeptide chain in space, provided that this course is well defined. The wording of the relevant sentences has been modified in the revised version to make this clear (page 6).

Line 776. 50 nM not 50 nm.

Author response: Thanks for catching the typo. It has been corrected in the revised version.

To summarise, this manuscript reports a very interesting observation but it fails to demonstrate if this represents an important step in the evolution of the GH1 family or just a curious artefact of ASR. More rigorous experiments are required to support the authors claims around protein dynamics, substrate specificity and affinity for heme.

Author response: We are, again, delighted that the reviewer finds that our manuscript reports a very interesting observation. We think that the additional and extensive experimental and computational work provided in the revised manuscript strongly support that heme binding is not a curious artefact of ASR and explicitly clarifies relevant issues related to substrate scope, the role of dynamics and the very high affinity of the ancestral glycosidase for heme.

REVIEWER COMMENTS

Reviewer #1 (Remarks to the Author):

The authors have carried out painstaking work to comply with numerous requests and suggestions from what seems to have been a demanding first round of reviewing. I believe that most issues have been clarified and the paper is now close to readiness for publication. I have only minor comments.

Possibly the authors mean:

Line 310 "We carried out ..."

Line 912: "Note the difference in the color bars ..."

Lines 914-5 Either "The numerical scale corresponds ..." or "The numerical scales correspond ..."

Line 1257: "similar to that observed"

Figure 7: The authors should acknowledge or cite any software used in the production of the panels of this Figure.

Line 1007: Does this mean "at 100 different intervals" or "at intervals of 100 generations", or something else?

Line 1267: Were periodic boundary conditions used? If so, how was the tessellation completed and was the octahedron a regular one - since I understand that regular octahedral boxes don't fill space without gaps, though an eight-faced hexagonal prism does?

Reviewer #2 (Remarks to the Author):

Thank you for addressing my concerns so thoroughly.

Small note:

Supplemental material no longer in numeric order

S20 – were the protein concentrations normalized? Are these replicates? What is the error? Heme loading can differ between protein preps.

Reviewer #3 (Remarks to the Author):

The authors have done a substantial amount of additional work to strengthen the evidence for their hypotheses. The revised version of the manuscript is significantly stronger in my opinion.

Reviewer #4 (Remarks to the Author):

One of my original reservations about this work was whether or not the observation of heme binding was a genuine part of GH1 evolution or simply an artefact of ASR. As such, I thought it was great that the authors tried to address this question by expressing several other ancestral and modern GH1 sequences with UV-Vis used to detect the presence of heme in protein samples (Figure S21).

However, the characterisation of these new ancestral proteins and the data collected is rudimentary at best and buried away in the SI. I have several concerns about this data, including:

1) The presence of heme in all these samples is inferred solely by the presence of a Soret band. It is not quantitated (no calibration curves) nor is it definitively identified by other techniques (e.g. MS).

2) All proteins except N98 and *S. degradans* GH1 showed barely-detectable levels of heme. If heme-binding was a true feature of GH1 evolution, wouldn't you expect substantial heme binding for the sequence nodes between them N72 and N98 (i.e. N73, N74, N75, N83)?

3) The proteins have only been purified by IMAC and a desalting column and there is no information provided regarding protein purity. How can the authors exclude the possibility that some of the heme detected here is from contaminating proteins or that it is due to non-specifically bound heme?

4) It is unclear if these other ancestral proteins are catalytically active or correctly folded. The authors should consider collecting some quick and easy data to address these points (kcat/Km by substrate depletion method or a simple single concentration reaction curve, CD etc.)

5) The Soret band for N98 and *S. degradans* GH1 are weaker and red-shifted than for N72. Indeed, many Soret bands for the other proteins appear red-shifted relative to N72. This really makes me concerned that a contaminant may be responsible for this signal.

Ultimately, the claim that heme-binding was a genuine step in GH1 evolution requires robust evidence with $n > 1$. I recommend MS analysis and heme Kd determination (by MST as has been done for N72) for each of these other node sequences. This shouldn't be too onerous given that the authors have already produced these proteins and established the MS and MST assays. This data should be placed in a main text figure, rather than the SI. I recommend using it to replace the horribly qualitative Figure 5, which should be relegated to the SI.

As the other referees and I have noted, the change in kcat/Km upon heme binding to N72 is very modest. The authors should tone-down their claims around rate-enhancement upon heme binding as this is not strongly supported by their data (e.g. L56-57 in Abstract: "Heme binding rigidifies this TIM-barrel and allosterically enhances catalysis" might be too bold a claim).

The authors have addressed all other concerns I raised in the first round of reviews to my satisfaction. However, as stated above, they really need to do more to nail-down heme binding to a few other nodes in the GH1 evolutionary tree... $n=1$ is not sufficient evidence to support the bold claim that heme binding is a general feature of ancestral or modern GH1 enzymes.

REVIEWER COMMENTS

Reviewer #1 (Remarks to the Author):

The authors have carried out painstaking work to comply with numerous requests and suggestions from what seems to have been a demanding first round of reviewing. I believe that most issues have been clarified and the paper is now close to readiness for publication.

RESPONSE: We are delighted that the reviewer finds that the first revised version already clarified most of the issues raised in the first round of reviewing.

I have only minor comments.

Possibly the authors mean:

Line 310 "We carried out ..."

Line 912: "Note the difference in the color bars ..."

Lines 914-5 Either "The numerical scale corresponds ..." or "The numerical scales correspond ..."

Line 1257: "similar to that observed"

RESPONSE: Thanks very much for catching these typos. They have been corrected in the revised version.

Figure 7: The authors should acknowledge or cite any software used in the production of the panels of this Figure.

RESPONSE: Figures displaying 3D-structures have been prepared using PyMOL. The 2D-interaction diagram of Figure 7A was prepared using LigPlot+. We acknowledge this in the revised version (page 35) and we provide the adequate reference.

Line 1007: Does this mean "at 100 different intervals" or "at intervals of 100 generations", or something else?

RESPONSE: Sorry for the ambiguous statement. It means "with samplings at intervals of 100 generations". It has been corrected in the revised version (page 30).

Line 1267: Were periodic boundary conditions used? If so, how was the tessellation completed and was the octahedron a regular one - since I understand that regular octahedral boxes don't fill space without gaps, though an eight-faced hexagonal prism does?

RESPONSE: Periodic boundary conditions (PBC) were indeed used, with a truncated octahedral box with a distance of 10 Å from the solute to the surface of the box. The box was then filled with TIP3P water molecules. The truncated octahedron can fill space without leaving any gaps, since our protein has a globular shape, a truncated octahedral box is the most suitable box shape to reduce the number of water molecules necessary to fill the box, which saves substantial computational time. This was generated using default AMBER settings, unfortunately we cannot find information about the tessellation AMBER uses for this in their documentation. We have updated the Methodology section ("Molecular dynamics simulations", page 36) to explicitly state that a truncated octahedral box was used, rather than just octahedral as originally stated.

Reviewer #2 (Remarks to the Author):

Thank you for addressing my concerns so thoroughly.

RESPONSE: We are delighted that the reviewer finds that his/her concerns have been thoroughly addressed.

Small note:

Supplemental material no longer in numeric order

RESPONSE: We have carefully checked the second revised version we are submitting and the numeric order in the Supplemental material follows the order in which the items (Supplemental tables and figures) are first mentioned in the main text. Please note that some of the items are mentioned several times in the main text and they may seem to be out of numeric order in the Supplementary Material if the first time they are mentioned was overlooked.

S20 – were the protein concentrations normalized? Are these replicates? What is the error? Heme loading can differ between protein preps.

RESPONSE: Yes, the spectra given in Figure S20 are normalized, since the y-axis shows extinction coefficient. The different molar extinction coefficients at the maximum of the protein band reflect different compositions in terms of aromatic amino acids (tryptophan in particular).

The figure does not show replicates and the reviewer is of course right that heme loading can differ between different preparations. However, we think that this is not an issue in the second revised version we are submitting now, because we have included a number of additional experiments that more clearly show heme binding to modern and ancestral glycosidases (see Figure 8 in the main text and Figure S23).

Reviewer #3 (Remarks to the Author):

The authors have done a substantial amount of additional work to strengthen the evidence for their hypotheses. The revised version of the manuscript is significantly stronger in my opinion.

RESPONSE: We are delighted that the reviewer finds that the large amount of additional work we carried out for the first revised version has substantially strengthened the manuscript.

Reviewer #4 (Remarks to the Author):

One of my original reservations about this work was whether or not the observation of heme binding was a genuine part of GH1 evolution or simply an artefact of ASR. As such, I thought it was great that the authors tried to address this question by expressing several other ancestral and modern GH1 sequences with UV-Vis used to detect the presence of heme in protein samples (Figure S21).

However, the characterisation of these new ancestral proteins and the data collected is rudimentary at best and buried away in the SI. I have several concerns about this data, including:

1) The presence of heme in all these samples is inferred solely by the presence of a Soret band. It is not quantitated (no calibration curves) nor is it definitively identified by other techniques (e.g. MS).

RESPONSE: In the revised version, we have included a number of additional *in vitro* experiments that clearly show heme binding to the modern and the ancestral glycosidases. The results of these experiments are collected in a new figure (Figure 8, see also Figure S23) and described in the main text (pages 12-13). In addition, the presence of heme in several of the ancestral protein samples resulting from the *in vitro* binding experiments has been confirmed by mass spectrometry (Figures S24, S25 and S26).

2) All proteins except N98 and *S. degradans* GH1 showed barely-detectable levels of heme. If heme-binding was a true feature of GH1 evolution, wouldn't you expect substantial heme binding for the sequence nodes between them N72 and N98 (i.e. N73, N74, N75, N83)?

RESPONSE: In the revised version, we report *in vitro* experiments that show amounts of bound heme that are much larger than those obtained in the "*in vivo*" experiments including the heme precursor in the culture medium. Please compare Figure S23 with the middle panel of Figure S21. Note that the intensity of the Soret band in the spectra of Figure S23 is comparable with the intensity of the protein band. Also, please compare, panels B and C in the new Figure 8 of the main text, which conveys the same information.

3) The proteins have only been purified by IMAC and a desalting column and there is no information provided regarding protein purity. How can the authors exclude the possibility that some of the heme detected here is from contaminating proteins or that it is due to non-specifically bound heme?

RESPONSE: Information regarding purity from SDS gel electrophoresis is now provided in the revised version (Figure S28). The fact that we detect large amounts of bound heme detected in the *in vitro* binding experiments we now report (Figure 8 and Figure S23) makes it highly unlikely that we are seeing at heme from contaminants or from non-specific binding.

4) It is unclear if these other ancestral proteins are catalytically active or correctly folded. The authors should consider collecting some quick and easy data to address these points (kcat/Km by substrate depletion method or a simple single concentration reaction curve, CD etc.)

RESPONSE: We have performed the experiments requested by the reviewer. CD spectra for the ancestral and modern proteins are now reported (Figure S29) and support that the proteins are correctly folded. Also, as suggested by the reviewer, we have performed single-concentration determinations of protein activity with two different substrates (Figure S22). The ancestral proteins are catalytically active and, in fact, the intermediate nodes (N73 to N100) display catalysis levels intermediate between the ancestral glycosidase at node N72 and the modern glycosidase from *Halothermothrix orenii*.

5) The Soret band for N98 and *S. degradans* GH1 are weaker and red-shifted than for N72. Indeed, many Soret bands for the other proteins appear red-shifted relative to N72. This really makes me concerned that a contaminant may be responsible for this signal.

RESPONSE: As noted above, the fact that we reveal large amounts of bound heme detected in the *in vitro* binding experiments (Figure 8 and Figure S23) makes it highly unlikely that we are seeing at heme contaminants or non-specific binding.

Ultimately, the claim that heme-binding was a genuine step in GH1 evolution requires robust evidence with $n > 1$. I recommend MS analysis and heme Kd determination (by MST as has been done for N72) for each of these other node sequences. This shouldn't be too onerous given that the authors have already produced these proteins and established the MS and MST assays. This data should be placed in a main text figure, rather than the SI. I recommend using it to replace the horribly qualitative Figure 5, which should be relegated to the SI.

RESPONSE: The additional experimental results reported in the second revised version provide robust evidence of heme binding with n now > 1 (11 GH1 proteins in total, including 4 modern proteins and 7 ancestral proteins). Mass spectrometry evidence for heme binding has been presented (Figures S24, S25 and S26). We have not performed Kd determination by MST on all the proteins because the expected evolutionary degradation of ancestral heme binding does not occur only at the level of binding strength. In fact, as our analyses of elution profiles from gel filtration chromatography show (Figure 8D and new text on pages 12-13), degradation is revealed by a trend towards decreased amount of heme-bound monomers and appearance of higher association states upon heme binding is observed in the evolutionary line leading to the modern glycosidase from *Halothermothrix orenii*.

As suggested by the reviewer, most of the new data are placed in a main text figure in the second revised version (Figure 8). However, we have deemed convenient to keep Figure 5. We agree that it is a qualitative figure, but, often, unexpected and shocking results are more easily accepted when they can be illustrated visually [as a well-known example, the fact that oxygen binding to haemoglobin causes a conformational change is more directly illustrated by the fact that reduced haemoglobin crystals can be seen to crack when exposed to oxygen (Perutz et al., Nature 4946:687-690, 1964)].

As the other referees and I have noted, the change in k_{cat}/K_M upon heme binding to N72 is very modest. The authors should tone-down their claims around rate-enhancement upon heme binding as this is not strongly supported by their data (e.g. L56-57 in Abstract: "Heme binding rigidifies this TIM-barrel and allosterically enhances catalysis" might be too bold a claim).

RESPONSE: We hope the reviewer is now willing to negotiate this point. First, even if there is minimal effect of heme binding on k_{cat}/K_M for the glucopyranoside substrate, there is always a substantial increase in k_{cat} and, it is important to note, technological applications of enzymes most often imply saturating substrate concentrations (i.e., k_{cat} may be more important than k_{cat}/K_M in a technological application scenario). Second, as we note in the manuscript, directed evolution can be trusted to enhance, fine-tune or even redesign a functionality, provided that a level of that functionality is available to start with. Even a low level of a new functionality thus opens up new possibilities for

protein engineering. We conclude, therefore, that our discovery of a new functionality in ancestral GH1 glycosidases (heme binding and allosteric modulation) should not be toned down.

The authors have addressed all other concerns I raised in the first round of reviews to my satisfaction. However, as stated above, they really need to do more to nail-down heme binding to a few other nodes in the GH1 evolutionary tree...n=1 is not sufficient evidence to support the bold claim that heme binding is a general feature of ancestral or modern GH1 enzymes.

RESPONSE: We are delighted that the reviewer finds that most of his/her concerns were satisfactorily addressed in the first revised version. We believe that the additional experimental studies included in the second revised strongly support the generality of heme binding to ancestral and modern GH1 glycosidases, as requested by the reviewer.

REVIEWERS' COMMENTS

Reviewer #1 (Remarks to the Author):

As I have already indicated with the previous version, I am happy with the revisions that the authors have implemented to the original manuscript.

Reviewer #4 (Remarks to the Author):

The revised manuscript has significantly strengthened the author's claim that heme binding is a general feature of ancestral GH-1 enzymes and I think the manuscript is now suitable for publication in Nat. Commun.

The authors and I still have differing views on the significance of the rate enhancement upon heme binding and the utility of this phenomenon for directed evolution experiments (to produce what and why, I am still unsure). However, this is just a matter of opinion and shouldn't get in the way of publishing what is now a large and significant body of work.

One minor issue that should be addressed prior to publication is the low quality of the gels in Figure S28. The lanes are wonky, the MW ladder is difficult to interpret and the protein samples themselves have a lot of contaminants in them.

Reviewer 4: "One minor issue that should be addressed prior to publication is the low quality of the gels in Figure S28. The lanes are wonky, the MW ladder is difficult to interpret and the protein samples themselves have a lot of contaminants in them"
RESPONSE: Gels of higher quality, showing a clear MW ladder, are included in Figure S28 of the final version.